# Substrate recognition and cryo-EM structure of the ribosome-bound TAC toxin of *Mycobacterium tuberculosis*

Moise Mansour [1,6], Emmanuel Giudice [2,6], Xibing Xu[1], Hatice Akarsu [3,5], Patricia Bordes[1], Valérie Guillet[4], Donna-Joe Bigot[1,4], Nawel Slama[1], Gaetano D'urso [2], Sophie Chat [2], Peter Redder [1], Laurent Falquet [3], Lionel Mourey [4], Reynald Gillet [2✉] & Pierre Genevaux [1✉]

Toxins of toxin-antitoxin systems use diverse mechanisms to control bacterial growth. Here, we focus on the deleterious toxin of the atypical tripartite toxin-antitoxin-chaperone (TAC) system of *Mycobacterium tuberculosis*, whose inhibition requires the concerted action of the antitoxin and its dedicated SecB-like chaperone. We show that the TAC toxin is a bona fide ribonuclease and identify exact cleavage sites in mRNA targets on a transcriptome-wide scale in vivo. mRNA cleavage by the toxin occurs after the second nucleotide of the ribosomal A-site codon during translation, with a strong preference for CCA codons in vivo. Finally, we report the cryo-EM structure of the ribosome-bound TAC toxin in the presence of native *M. tuberculosis* cspA mRNA, revealing the specific mechanism by which the TAC toxin interacts with the ribosome and the tRNA in the P-site to cleave its mRNA target.

[1] Laboratoire de Microbiologie et de Génétique Moléculaires, Centre de Biologie Intégrative (CBI), Université de Toulouse, CNRS, UPS, Toulouse, France. [2] Institut de Génétique et Développement de Rennes (IGDR), UMR6290, Université de Rennes, CNRS, Rennes, France. [3] Department of Biology, University of Fribourg & Swiss Institute of Bioinformatics, Fribourg, Switzerland. [4] Institut de Pharmacologie et de Biologie Structurale, IPBS, Université de Toulouse, CNRS, UPS, Toulouse, France. [5] Present address: Institute of Veterinary Bacteriology, University of Bern, Bern, Switzerland. [6] These authors contributed equally: Moise Mansour, Emmanuel Giudice. ✉email: reynald.gillet@univ-rennes1.fr; pierre.genevaux@univ-tlse3.fr

Toxin–antitoxins (TA) are two-component systems composed of a poisonous toxin and a counteracting antitoxin[1–3]. Toxins generally employ very diverse and elegant strategies to inhibit bacterial growth, mainly targeting essential processes or structures, including translation, replication, or membranes[2,4–12]. Although the specific conditions that lead to toxin activation in vivo remain largely unknown[13], TA systems have been involved in the maintenance of chromosomes, plasmids, or other genetic mobile elements, in the defense against phage infection through a process known as abortive infection and in some cases, in antibiotic persistence in vivo in infectious models and in bacterial virulence[14–20].

*Mycobacterium tuberculosis*, the bacterium responsible for human tuberculosis, encodes for more than 86 TA systems, covering large sets of conserved TA families, including VapBC (50 pairs), MazEF/PemIK (11), MenTA (4), HigBA (3), RelBE (3), ParDE (2), Res-Xre (2), DarTG (1), PezAT (1), and RHH-GNAT (1), as well as other less conserved or uncharacterized TA pairs[21,22]. Many *M. tuberculosis* toxins inhibit bacterial growth when overexpressed and TA operons are often induced under relevant stresses, including hypoxia, macrophage engulfment, or antibiotics[23,24]. This suggests that stress-activated toxins could potentially modulate the growth of *M. tuberculosis* and contribute to its success as a major human pathogen[22,24]. Yet, except for *ΔvapC22*, *ΔvapBC3/4/11* and *ΔmazF3/6/9* mutants that were shown to be impaired in host infection, almost nothing is known about the role of TA modules in the biology of *M. tuberculosis*[25–27].

Rv1955-Rv1956 (HigBA^TAC, also named HigBA1), Rv2022c-Rv2021c (HigBA2), and Rv3182-Rv3183 (HigBA3) were originally annotated as HigBA-like TA pairs due to their less common inverted genetic organization, i.e., the toxin encoded first, similar to the HigBA (host inhibition of growth) module found on the Rts1 plasmid of *Proteus vulgaris*[28]. Such a module was shown to contain a RNase toxin (HigB) that possesses a similar fold as the RelE[29] toxin of *Escherichia coli* and a HTH-Xre DNA-binding domain containing antitoxin (HigA) (Fig. 1a, b). The HigBA^TAC pair is part of the atypical tripartite TAC (toxin–antitoxin–chaperone) system in which efficient inhibition of the HigB^TAC toxin (also named HigB1) by the antitoxin (HigA^TAC, also named HigA1) relies on the molecular chaperone Rv1957 (SecB^TA) encoded as the third gene of the *TAC* (*higB-higA-Rv1957*) operon[30,31]. The TAC chaperone, which is homologous to the export chaperone SecB of *E. coli*[32,33], specifically interacts with the C-terminal chaperone-addiction region of the antitoxin to prevent its aggregation and degradation, and to facilitate subsequent inhibition of the toxin by the antitoxin[34–36]. Although transcription of *TAC* is induced under relevant stresses for *M. tuberculosis*, including nutritional starvation, hypoxia, antibiotics treatment, and drug-induced persistence[22,23,37–39], very little is known about its possible involvement in *M. tuberculosis* physiology and virulence[40]. Deletion of the HigA^TAC antitoxin is lethal, possibly due to the synthesis of free active toxin[39]. Accordingly, overexpression of the HigB^TAC is toxic in *M. tuberculosis*, *Mycobacterium smegmatis*, *Mycobacterium marinum*, and *E. coli*[30,35,41], and was shown to affect the abundance of a subset of transcripts related to iron and zinc homeostasis[41]. In contrast with the TAC system, nearly nothing is known about the putative function and activity of HigBA2 and HigBA3, except that ectopic expression of the HigB3 toxin does not affect *M. smegmatis* growth and that neither *higA2* nor *higA3* are essential for *M. tuberculosis*[23,42].

In this work, we investigate the in vivo targets and the molecular mechanism of HigB-like toxins of *M. tuberculosis*, mainly focusing on the TAC toxin. We find that among the three toxins, only HigB^TAC exhibits robust toxicity and efficiently inhibits translation. In addition, we show that HigB^TAC functions as a bona fide RNase that specifically cleaves mRNA after the second nucleotide of A-site codons (mainly CCA codons) during translation. Finally, we solve both the X-ray structure of HigB^TAC alone and the cryo-EM structure of the ribosome-bound HigB^TAC in the presence of its native *M. tuberculosis cspA* mRNA. The specific mechanism by which HigB^TAC interacts with the ribosome and the tRNA at the P-site to cleave its mRNA target is discussed.

## Results

### HigB^TAC, but not HigB2 and HigB3, is toxic and inhibits protein synthesis.

In order to assess the toxicity of the three *M. tuberculosis* HigB-like toxins in comparable expression systems, the toxins were cloned and expressed in *M. smegmatis* under the control of an anhydrotetracycline (ATc) inducible promoter on an integrative plasmid. *M. smegmatis* does not have endogenous HigA antitoxin, which is very convenient to study the impact of solitary HigB toxins. Under these conditions, we found that the expression of HigB^TAC induces a robust toxic phenotype (Fig. 1c). In addition, alanine substitution of residue K95 of HigB^TAC, which possibly corresponds to the active site residue R81 of *E. coli* RelE[29], fully inhibited toxicity in vivo in *M. smegmatis* (Fig. 1c and Supplementary Fig. 1a). This strongly suggests that HigB^TAC toxicity was due to its putative RNase activity. Note that the K95A substitution that leads to the inactive toxin was used as non-toxic HigB^TAC control for further in vivo and in vitro analysis. In contrast, no toxicity was observed for HigB2 or HigB3 under the same conditions. In a similar manner, no toxicity was observed when these toxins were expressed in *M. smegmatis* from a high copy number plasmid under the control of an acetamide-inducible promoter (Supplementary Fig. 1a) or in *E. coli* following expression from an arabinose-inducible promoter (Supplementary Fig. 1b). In addition, expression of HigB toxins could only be detected on SDS-PAGE when expressed from the strong T7 promoter of pET-vector in *E. coli*. In this case, both HigB2 and HigB3 toxins showed higher expression levels than HigB^TAC (Supplementary Fig. 1c). The lack of toxicity of HigB2 and HigB3 is further supported by the fact that their respective antitoxins are not essential for *M. tuberculosis* growth[42].

To further investigate the difference in growth inhibition observed between the three toxins, the impact of purified HigB^TAC, HigB^TAC[K95A], HigB2, or HigB3 on protein synthesis was monitored in vitro by following the expression of the model protein GFP in a coupled transcription/translation assay[34]. In agreement with the toxicity data, HigB^TAC efficiently blocked GFP synthesis while HigB2, HigB3, and the HigB^TAC[K95A] mutant did not (Fig. 1d). Note that, at a high concentration of toxins (i.e., 20 times higher than the one at which HigB^TAC wild-type efficiently inhibits translation), we do observe a decrease in GFP synthesis in the presence of HigB2, HigB3, or HigB^TAC[K95A]. In contrast, the addition of unrelated proteins at the same concentration did not show such a marked effect on GFP synthesis (Supplementary Fig. 1d). This suggests that these toxins might have a lower affinity for the ribosome or be catalytically inactive but still able to interact with the ribosome and/or the mRNA, and somehow interfere with translation under our in vitro conditions.

### Genome-wide identification of toxin targets and cleavage preference.

A nEMOTE (nonphosphorylated exact mapping of transcriptome ends) approach was used to identify HigB toxin substrates and recognition sequences in vivo for further in vitro characterization. This qualitative method, which allows the exact mapping of 5'-OH cleaved sites in mRNA in vivo on a transcriptome-wide scale

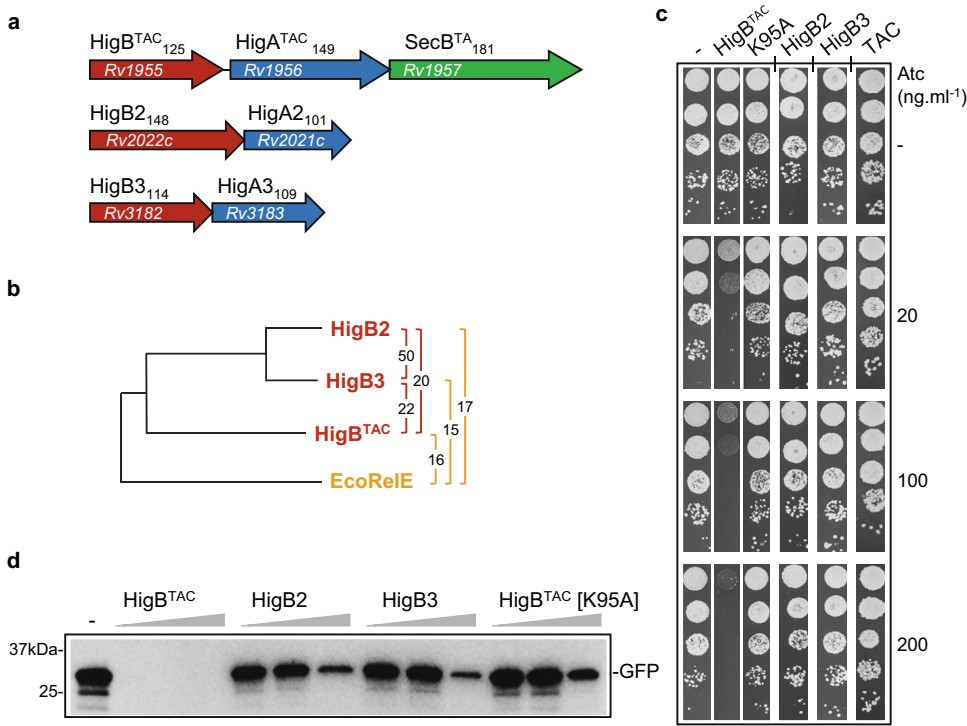

**Fig. 1 Impact of HigB toxins on growth and protein synthesis. a** Schematic representation of the three HigBA-like systems of *M. tuberculosis*. Toxins are shown in red, antitoxins in blue and the chaperone of the TAC system (SecB$^{TA}$) in green. The amino acid length of each protein is given as a subscript and the locus tag for each gene within color arrows. **b** Phylogenetic tree of *M. tuberculosis* HigB toxins (red) and *E. coli* RelE (orange). The percentage of sequence identity between pairs of toxins is shown on the right (colored brackets). **c** Expression of HigB toxins in vivo. *M. smegmatis* transformed with pGMC-vector (-), HigB$^{TAC}$, HigB$^{TAC}$ [K95A], HigB2 or HigB3 or TAC (HigB/HigA/SecB)$^{TAC}$ were serial diluted and spotted on LB agar plates supplemented with or without Atc inducer at the indicated concentration. Plates were incubated for 3 days at 37 °C. **d** In vitro transcription/translation reactions assessing the synthesis of GFP protein in the absence (-) or presence of increasing concentrations of HigB toxins, (0.3, 3, and 6 μM). Samples were separated by SDS-PAGE and revealed by western blot using an anti-GFP antibody. Representative results of three independent experiments are shown.

has been successfully used in *Caulobacter crescentus* and *Staphylococcus aureus*[43–45]. The nEMOTE analyzes were performed with HigB$^{TAC}$, HigB$^{TAC}$[K95A] as negative control, HigB2, and HigB3 expressed in both *M. smegmatis* WT and Δ*rnJ*. The Δ*rnJ* mutant does not produce ribonuclease J and was used to minimize degradation of cleaved toxin targets and thus potentially increase the robustness of the signal of the preferred cleavage sites[46]. Although HigB2 and HigB3 were not toxic, we hypothesized that they might still exhibit some weak RNase activity. Expression of the toxins was induced with ATc for 3 hours (or additionally for 24 hours in the case of HigB2 and HigB3) at mid-log phase, whereupon total RNAs were extracted and analyzed with nEMOTE (see "Methods"). Cleavage sites for each condition are presented in Supplementary Data 1 and the identified target genes are summarized in Fig. 2a and Supplementary Fig. 2a. For this analysis, we considered that a cleavage site for a given toxin should be found at least seven times in one condition and at least in two out of the six replicates to be counted as a bona fide target site in vivo (see Bioinformatic analysis in the Methods section). Remarkably, ectopic expression of HigB$^{TAC}$ led to the identification of 24 unique cleavage sites located within 22 different RNA targets (Fig. 2a). Eight of these cleaved coding regions of essential proteins are conserved in *M. tuberculosis* and affect different cellular processes, including translation, replication, protein folding, and metabolism (Supplementary Fig. 2a). Additionally, four cleavage sites were identified in previously non-annotated transcripts (named nat1 to nat4). We found a strong overlap among the identified target sites from the WT and Δ*rnJ* mutant, with 15 out of 24 found in both strains. Strikingly, all the cleavages identified in annotated open reading frames occurred between the second and

the third nucleotide of the codons, suggesting that HigB$^{TAC}$ might preferentially cleave mRNA during translation. Alignment of all the sequences flanking the cleavage sites revealed a clear preference for CCA codons (encoding a proline), with cleavage occurring between the C and A nucleotides (Fig. 2b and Supplementary Fig. 2b). In sharp contrast, expression of HigB$^{TAC}$[K95A] only led to a single potential hit in a non-annotated transcript (nat5) that appeared with low frequency (22 cuts in 2 replicates), which could be a false positive caused by the background noise or a very weak RNase activity (Supplementary Fig. 2a, c). Similarly, only 3 and 1 hits were identified for HigB2 and HigB3, respectively. Such hits were only detected after prolonged exposure to the toxins (24 h instead of 3 h in the case of HigB$^{TAC}$) in the wild-type strain and with a low frequency (Supplementary Data 1 and Supplementary Fig. 2c). These data are in line with the lack of toxicity observed in vivo.

## Ribosome dependence and cleavage preference by TAC toxin in vitro.
The nEMOTE data highlighted several potential substrates of HigB$^{TAC}$, including the frequently cleaved *cspA* and *groES* transcripts, whose cleavage sites are conserved in both *M. smegmatis* and *M. tuberculosis* (Fig. 2a and Supplementary Fig. 2a). In agreement with the in vivo data, we found that both *cspA* and *groES* of *M. tuberculosis* were also bona fide HigB$^{TAC}$ substrates in vitro, as judged by HigB$^{TAC}$ ability to inhibit *M. tuberculosis* CspA (Fig. 3a) and GroES (Supplementary Fig. 3a) synthesis by *E. coli* ribosomes. The *cspA* mRNA, which has only one HigB$^{TAC}$ cleavage site in its sequence, i.e., a CCA codon at the amino acid position 2, was selected as a model substrate transcript to further study HigB$^{TAC}$

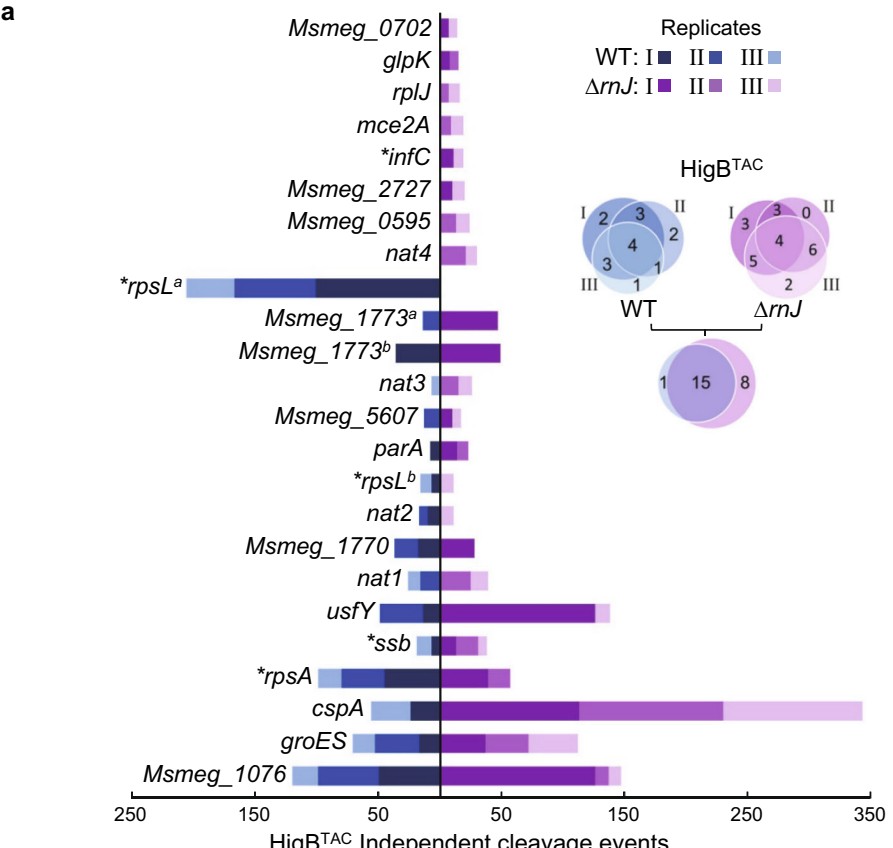

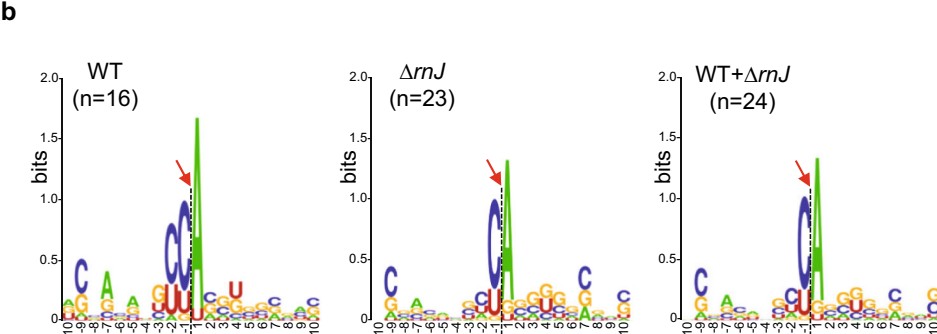

**Fig. 2 HigB$^{TAC}$ targets and recognition sequences in vivo. a** HigB$^{TAC}$ targets identified in vivo by nEMOTE following 3 h expression in *M. smegmatis* WT (on the left; shade of blue; replicates I, II, and III) and Δ*rnJ* (on the right; shade of violet; replicates I, II, and III). Target overlaps between the replicates and between the strains are represented by Venn diagrams. Bar length represents the number of independently observed cleavage events for each unique target site (*x* axis). The name of the gene of the cleaved mRNA is given on the left. The number of cleavages identified in each replicate for the WT (shade of blue; replicates I, II, and III) and the Δ*rnJ* mutant (shade of violet; replicates I, II, and III) is shown using the indicated color code. (*) indicates essential genes in both *M. smegmatis* and *M. tuberculosis*. **b** HigB$^{TAC}$ preferred sequence motif. Logoplots showing HigB$^{TAC}$ preferred motif obtained with the unique cleavage sequences identified for the WT (*n* = 16), the Δ*rnJ* mutant (*n* = 23) or both WT + Δ*rnJ* (*n* = 24). The *x* axis represents the 10 nucleotides upstream and downstream of the cleavage site that is located by the position between −1 and 1 (red arrow and dash line), and the default label for the *y* axis is bits.

mode of action and motif preference in vitro. To confirm that inhibition of CspA synthesis was reflecting *cspA* mRNA cleavage, we performed primer extension experiments to monitor *cspA* transcript during translation in the cell-free coupled transcription/translation system (Supplementary Fig. 3b) and showed that *cspA* transcripts were indeed cleaved by HigB$^{TAC}$ (Fig. 3b) and not by the inactive HigB$^{TAC}$[K95A] substitution (Supplementary Fig. 3c). Remarkably, we found that *cspA* cleavage by HigB$^{TAC}$ occurred only in the presence of the ribosome (Fig. 3b) and that out of frame CCA motifs in *cspA* (OOF1 and OOF2) were not recognized by HigB$^{TAC}$, demonstrating that mRNA cleavage by

the TAC toxin is ribosome-dependent. Furthermore, although the CCA codon at the second position was the most robustly processed by HigB$^{TAC}$, we found that relocating CCA codon along the *cspA* transcript (with the native CCA at position 2 replaced by CCG also encoding a proline) results in cleavage at the new CCA sites, indicating that HigB$^{TAC}$ is capable of cleaving mRNA during translation elongation (Supplementary Fig. 4). This is in agreement with the nEMOTE data that identified HigB$^{TAC}$ cleavages along different transcripts (Supplementary Fig. 2a).

The in vivo nEMOTE data strongly suggest that HigB$^{TAC}$ cleaves between the second and the third nucleotide within codons,

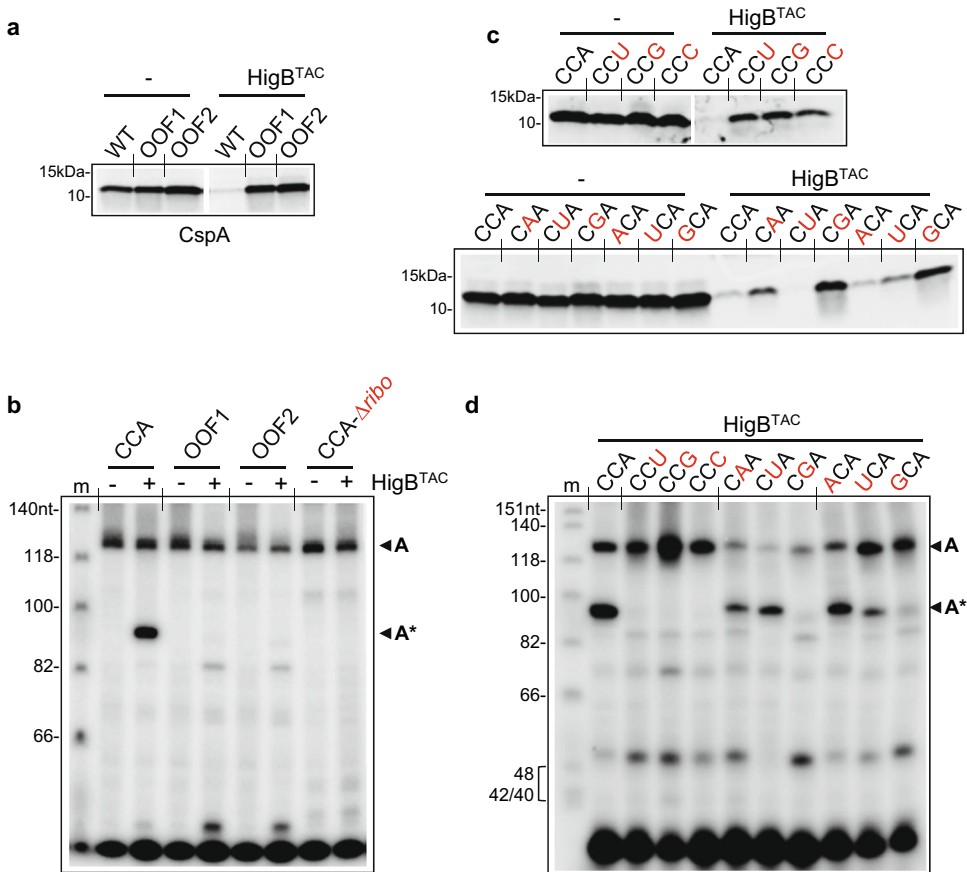

**Fig. 3 HigB[TAC] cleaves *cspA* mRNA at CCA codon during translation in vitro. a** HigB[TAC] inhibits synthesis of CspA wild-type (WT with codon CCA at Pro2) but not of CspA with +1 (OOF1) and +2 (OOF2) out of frame CCA motifs. CspA WT, OOF1, and OOF2 were independently expressed in a cell-free translation system with or without HigB[TAC] (1.5 μM) as described in Fig. 1d. CspA translation products were labeled with [35S]methionine and reactions were performed for 1 h 30 min at 37 °C. After translation, samples were separated on SDS–PAGE and visualized by phosphorimager. **b** Cleavage of *cspA* wild-type is ribosome-dependent. *cspA* wild-type (CCA), OOF1, and OOF2 were independently expressed in a cell-free translation system for 2 h with or without HigB[TAC] (1.5 μM). RNA was extracted and subjected to a primer extension with [32P]-labeled *cspA* primer. In parallel, *cspA* mRNA was incubated for the same time with or without HigB[TAC] (1.5 μM) in the absence of ribosomes (CCA-Δ*ribo*), and also used for primer extension. The obtained labeled cDNAs were separated on denaturing urea-polyacrylamide gel and revealed by autoradiography. Arrows show the uncleaved (A, 126 nt) and cleaved (A*, 95 nt) *cspA*, and (m) stands for molecular ladder. Mutations in the CCA codon prevent both inhibition of CspA synthesis (**c**) and cleavage (**d**) of *cspA* by HigB[TAC] in vitro. *cspA* wild type (CCA) and its mutant derivatives (mutations depicted in red) were independently expressed in a cell-free translation system with or without HigB[TAC] (1.5 μM) and analyzed as described in panel (**a**) for CspA protein synthesis and in panel (**b**) for *cspA* cleavage. Representative results of triplicate experiments are shown.

with a preference for adenosine at the third position and with CCA codons being the most represented targets (Supplementary Fig. 2b). We next engineered mutations at each position of the CCA codon of *cspA* and tested them for translation inhibition and cleavage by HigB[TAC] in vitro (Fig. 3c, d and Supplementary Fig. 3d, e). Strikingly, our data show that the adenosine at the third position is indeed critical for cleavage by HigB[TAC], as all of the mutations at this position efficiently prevented both inhibition of protein synthesis and cleavage by the toxin. In contrast, cytosines at the first and second positions were generally less sensitive to certain mutations. Indeed, while both positions were severely impacted when mutated for guanine, other mutations led to a more minor (i.e., UCA and CAA) or no (i.e., ACA and CUA) detectable impact. These data are in strong agreement with the cleavage sites identified in vivo on a transcriptome-wide scale (Supplementary Fig. 2a) and demonstrate that the TAC toxin shows substantial substrate preference.

**Overall structures of isolated and 70S ribosome-bound HigB[TAC].** We solved the structure of HigB[TAC][K95A] in complex with the *E. coli* 70 S ribosome, fMet-tRNA[fMet] and native *M. tuberculosis*

*cspA* mRNA by single-particle cryo-EM, with an average resolution of 3.64 Å (Fig. 4a and Supplementary Fig. 5a–d). Note that exploitable cryo-EM data were obtained only when a high concentration of inactive toxin over ribosomes was used in the presence of full-length *cspA* mRNA (100 toxins for 1 ribosome, 2 tRNAs and 2 mRNAs; see "Methods" section). Interestingly, while the sample presented a certain heterogeneity with 70 S in classical or ratcheted conformation and with tRNA in the P- and/or E-site, the toxin was only observed on non-ratcheted ribosome with a tRNA in the P-site. This suggests that binding of the toxin to the ribosome relies on the presence of a tRNA in the P-site and that the toxin might be rapidly released upon ratcheting. Overall, our cryo-EM structure shows that HigB[TAC] occupies the ribosomal A-site and makes interactions with the 16S (h18, h30-31, h34, and h44) and the 23S (H69) rRNA, as well as with the P-site tRNA[fMet], the ribosomal protein uS13 and *cspA* mRNA (Fig. 4a, c). The fMet-tRNA[fMet] is bound to the AUG initiation codon in the P-site and the CCA codon (Pro 2) of *cspA* lies in the A site, in the vicinity of the toxin, which is in agreement with the in vivo and in vitro cleavages obtained at this position.

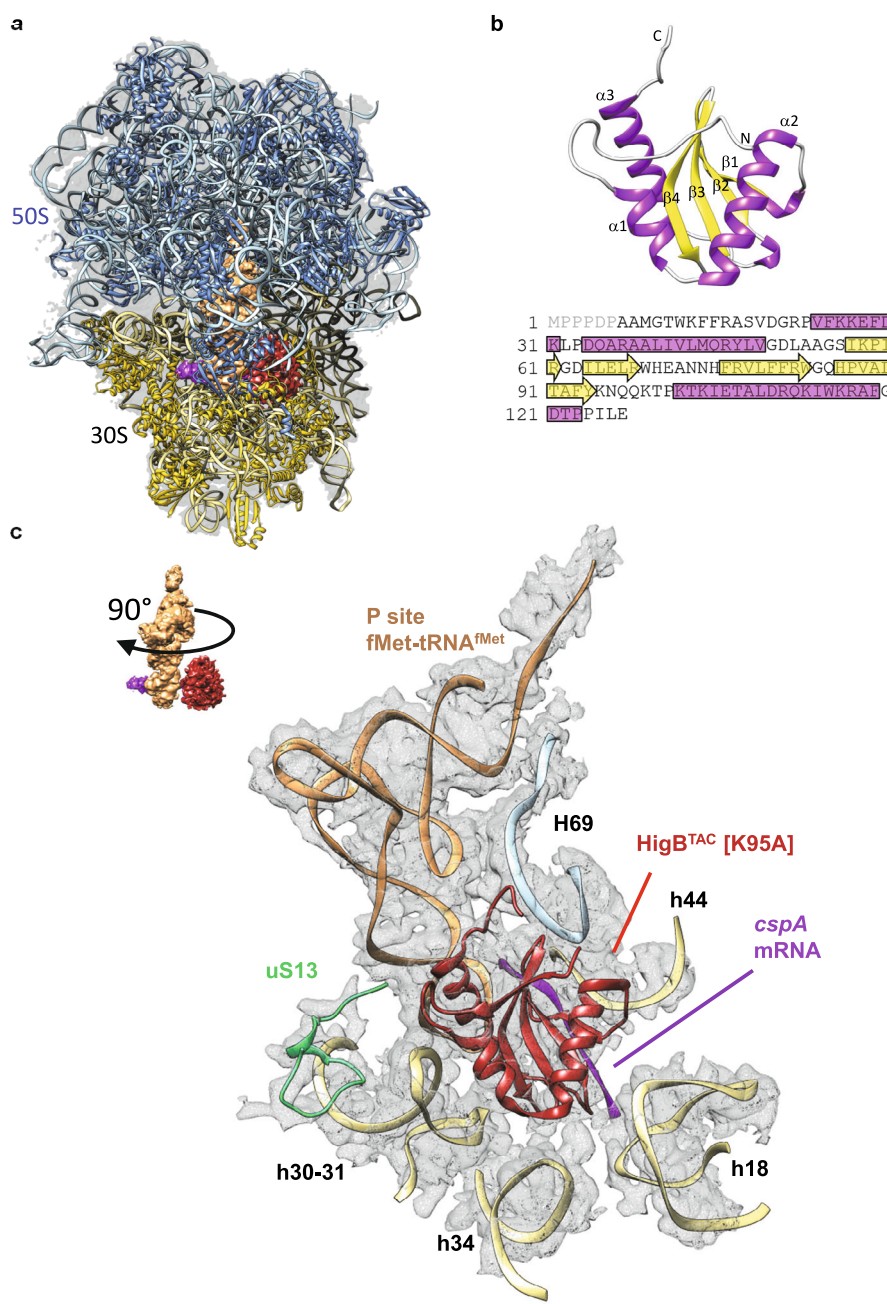

**Fig. 4 Structure of the ribosome-associated TAC toxin with its native *cspA* substrate. a** Electron density map of the complex (transparent gray), showing HigB[TAC] [K95A] (red), its target *cspA* mRNA (purple), fMet-tRNA[fMet] (orange), and the atomic models for the 50S (blue) and 30S (yellow) ribosomal subunits. **b** Crystal structure of HigB[TAC] [K95A]. The N-terminus, C-terminus, and the secondary structure elements are indicated. The sequence of the toxin is shown underneath with the secondary structure element highlighted (α-helix in pink, β-strand in yellow). The first six residues (light gray) are not visible in the crystal structure. **c** Close-up of the interactions between HigB[TAC] [K95A], the *cspA* mRNA, and the translating ribosome, rotated by 90° with respect to the structure in **a**. The HigB[TAC] [K95A] is red; the P-site fMet-tRNA[fMet] is orange; the *cspA* mRNA is purple; the uS13 ribosomal protein is green; the 16S rRNA is light yellow; the 23S rRNA is light blue; and the cryo-electron density map is a gray mesh. For clarity, 16S and 23S α-helices are labeled.

In parallel, the crystal structure of isolated monomeric HigB[TAC][K95A] was obtained at a 1.9 Å resolution and used to compare possible conformational changes that may occur upon association with the ribosome (Fig. 4b and Supplementary Fig. 5e, f). HigB[TAC] has a similar compact fold as other RelE homologs, with four antiparallel β-strands flanked by three α-helices[29,47,48]. In contrast with *E. coli* RelE[29] or *P. vulgaris* HigB[49], HigB[TAC] possesses a long C-terminal helix α3 (Fig. 4b and Supplementary Fig 6). Such long C-terminal helix is found in other closely related

Gp49-like toxins, including *S. pneumoniae* HigB[47] and *V. cholera* HigB2[48]. Both cryo-EM and crystal structures of HigB[TAC][K95A] were very similar, with an rmsd value of 1.9 Å, indicating that little conformational changes occur upon binding to the ribosome, except for the loop between β2 and β3 (residues 68 to 75), α1 and α2 (residues 31 to 35), β4 and α3 (residues 95 to 101), and the beginning of β1 (residues 54 to 56) that fold to interact with the 16S rRNA decoding site and the target mRNA. Notably, these regions include most of the proposed catalytic

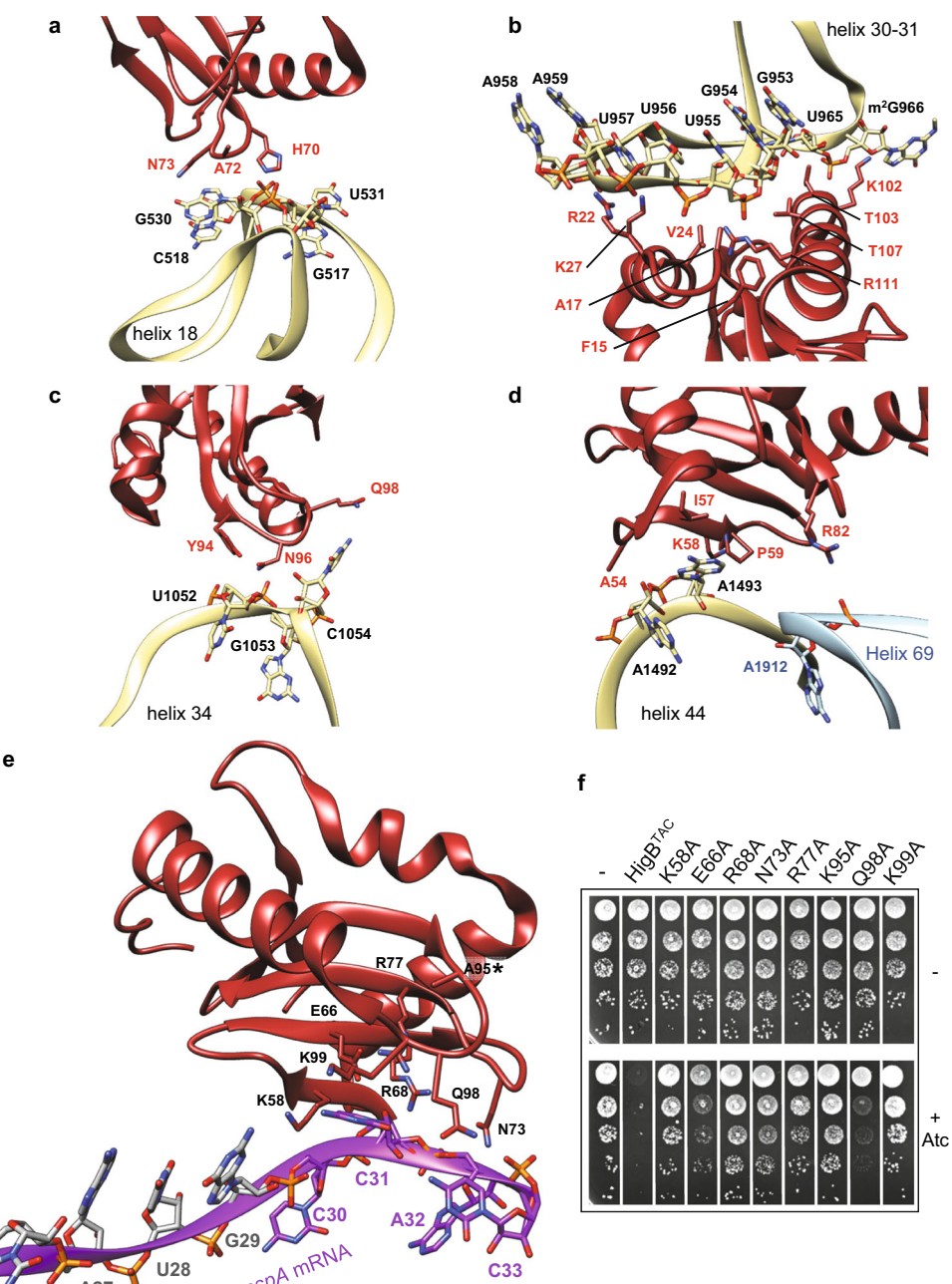

**Fig. 5 HigB$^{TAC}$ interactions with the ribosome and with *cspA* mRNA.** Interaction between HigB$^{TAC}$ [K95A] and: **a** helix 18 of 16S rRNA; **b** the region delimited by helices 30 and 31 of 16S rRNA; **c** helix 34 of 16S rRNA; and **d** helix 44 of 16S rRNA and helix 69 of 23S mRNA. **e** Interactions between the HigB$^{TAC}$ [K95A] catalytic site and the CCA codon of the *cspA* mRNA. In all figures, HigB$^{TAC}$ [K95A] is red, the *cspA* mRNA is purple, the 16S rRNA is light yellow, and the 23S rRNA is light blue. Residues and nucleotides within 4 Å of each other are indicated. **f** *M. smegmatis* transformed with pGMC-vector (−), HigB$^{TAC}$ wild-type or its mutant derivatives were serial diluted, spotted on LB streptomycin agar plates with or without anhydrotetracycline (Atc ng/mL) inducer and incubated 3 days at 37 °C. Representative results of three independent experiments are shown.

residues of HigB$^{TAC}$ (i.e., E66, R68, N73, K95, F93, Q98, and K99), suggesting that such differences could have some functional significance.

**HigB$^{TAC}$ interaction with 16S and 23S rRNA.** The 16S rRNA appears as the main contact site for HigB$^{TAC}$ on the ribosome (Fig. 5a–d). The interface engages more than twenty amino acid residues of HigB$^{TAC}$, covering a large surface of the toxin and interacting with the regions of the rRNA that include helices h18, h30-31, h34, h32, and h44 (Supplementary Fig. 7). Such

interactions are very similar to the ones observed for *E. coli* RelE and YoeB, and *P. vulgaris* HigB toxins[29,50,51]. A comparison of contact areas between rRNA and known ribosome-bound toxins is detailed in Supplementary Fig. 7. The main contact of HigB$^{TAC}$ with the 16S rRNA involves amino acid residues of its N-terminal region and of helices α1 and α3, as well as nucleotides at the junction between rRNA helices h30 and h31 (Fig. 5b). Using a cut-off distance of 3 Å, we considered that important residues include K27 that interacts with the O4 of U957 and R22 and R111 that are within hydrogen-bonding distance of the phosphate groups of A959 and U955, respectively. These residues may

provide key interaction for the correct positioning of HigB[TAC] into the A site. Important interactions are also provided by histidine 70 (located between β2 and β3), which contacts the phosphate group of C518 (Fig. 5a), and by residue N96 (located between β4 and α3) with G1053 (Fig. 5c). In addition, HigB[TAC] interacts with the decoding center, with residues A72 and N73 (located in the loop between β2 and β3) both interacting with G530 (Fig. 5a), and residues I57, K58, and P59 of β1 contacting A1493 (Fig. 5d). As a result, A1493 adopts an intermediate state between its fully flipped position observed during cognate mRNA-tRNA pairing and its position inside h44 observed when the A-site is vacant. A1492 remains mostly inside h44 while G530 is in a *syn* conformation, like in vacant 30S subunits. Together these results are consistent with what was previously observed for the pre-cleavage complex of *P. vulgaris* HigB on the ribosome[52] but differ from what was described for (i) RelE[29], where A1492 replaces A1493 outside of h44 and (ii) YoeB[53], where both A1492 and A1493 are fully flipped out (Supplementary Fig. 7). No major contact was found between HigB[TAC] and the 50S subunit, except residue R82, which sits within a 4 Å distance of residue A1912 of H69 (Fig. 5d).

**HigB[TAC] catalytic site and interaction with *cspA* mRNA.** In agreement with the in vivo and in vitro data, the CCA motif of native *cspA* transcript is directly facing HigB[TAC] in the ribosomal A-site. While the HigB[TAC]:*cspA* interface has an estimated local resolution of 3.3 Å, the densities are not very well defined and we were not able to rigorously orientate the nucleobases of C30 and A32, suggesting that interaction between the toxin and the CCA motif might be dynamic, at least in the case of HigB[TAC][K95A]. Yet, we could identify with confidence several key features of *cspA* and HigB[TAC] interaction (Fig. 5e). Residues K58, E66, R68, N73, R77, Q98, and K99 are within a 4 Å distance of *cspA* mRNA nucleotides C30, C31, A32 (CCA codon) and C33. Together with the mutated residue K95, these residues may form the catalytic pocket of HigB[TAC] that pulls the mRNA from its canonical position in the A-site, and surround the nucleotide C31. Based on available structures, residues R68, R77, and K95 of HigB[TAC] may correspond to the K54, R61, and R81 catalytic residues in RelE, and to R68, R73, and K92 in HigB from *S. pneumoniae* (Supplementary Fig. 6). In addition, residue K58 possibly corresponds to residues K57 in *P. vulgaris* HigB and K49 in YoeB that interact with the second nucleotide of the A-site mRNA codon[49]. Residue E66 in HigB[TAC] is conserved in *S. pneumoniae* HigB (residue E66) where it was proposed to be part of the catalytic site[47]. Alanine substitution of this residue partially inhibits HigB[TAC] activity in vivo (Fig. 5f and Supplementary Fig. 8), suggesting that E66 could indeed contribute to catalysis[49,51]. Amino acid F93 in HigB[TAC] corresponds to core residues F90 in *S. pneumoniae* HigB and Y82 in *V. cholerae* HigB2[48]. While Y82 was proposed to be involved in mRNA positioning[48], F93 is not in close contact with the mRNA in our structure. However, its current orientation suggests that it could be required to correctly position the proposed catalytic residues. More work will be needed to clarify the role of this residue. As observed for K95A, alanine substitutions of the positively charged residues K58, R68, R77, and K99 inhibited HigB[TAC] toxicity (Fig. 5f and Supplementary Fig. 8), highlighting their importance for HigB[TAC] activity in vivo. In addition, the N73A substitution had a strong effect on HigB[TAC] toxicity, while Q98A showed only a mild effect (Fig. 5f). Both Q98 and N73 point towards the adenosine at the third position of the CCA codon, which is critical for the mRNA recognition by HigB[TAC] (Fig. 3d). Remarkably, while N73 has no detectable equivalent in closely related toxins of *S. pneumoniae* and *V. cholerae*[47,48], it could have a similar role as residue N71 of *P.*

*vulgaris* HigB, which is in the vicinity of the third nucleotide of the A-site codon and contributes to the toxin sequence specificity[49]. Although *P. vulgaris* HigB was proposed to select for adenosine at the 3rd position of the A-site codon through a *trans* Watson–Crick–Hoogsteen interaction with 16S rRNA C1054[52], our structure did not reveal any base pairing between the *cspA* mRNA and C1054.

**HigB[TAC] α3 interacts with P-site fMet-tRNA[fMet] and S13 ribosomal protein.** The HigB[TAC] long C-terminal α3 (Figs. 4 and 6a) is found in HigB of *S. pneumoniae* and HigB2 of *V. cholera*, but not in *E. coli* RelE, YoeB, and *P. vulgaris* HigB (Fig. 6b and Supplementary Figs. 6 and 9). Our cryo-EM structure reveals that upon binding to the ribosome, one exposed face of α3 lies along the P-site fMet-tRNA[fMet]. As a result, HigB[TAC] is more deeply embedded in the A site than any other ribosome-dependent toxin described so far. Noticeably, positively charged residues K113 and R117 interact with the tRNA nucleotides G30 and C29, and R117 is also in close contact with residue I116 in the C-terminal tail of the ribosomal protein uS13 (Fig. 6a). This suggests that these residues may play an important role for HigB[TAC] activity. Accordingly, single alanine substitution of either K113 or R117 affected HigB[TAC] toxicity both in *M. smegmatis* and in *E. coli*, with K113A showing the strongest inhibitory effect (Fig. 6c and Supplementary Fig. 8). In contrast, alanine substitutions of the exposed negatively charged residues E106 and D110 of α3 did not inhibit HigB[TAC] toxicity. In addition to α3, interaction with the P-site fMet-tRNA[fMet] is also reinforced by residue R61, located at the end of strand β1, which lies within hydrogen bonding distances of the phosphate group between the fMet-tRNA[fMet] nucleotides A36 and U37. Finally, we purified HigB[TAC] [R61A] and HigB[TAC] [K113A] and tested their activity in vitro. Our data show that indeed both mutations abolished HigB[TAC] ability to inhibit CspA synthesis and severely affect *cspA* mRNA cleavage in vitro (Fig. 6d, e). Together these data suggest that the interactions between HigB[TAC] and the P-site tRNA are critical for the toxin activity.

**Discussion**

This work shows that the TAC toxin is a bona fide ribosome-dependent RNase that binds the ribosome A-site to cleave its mRNA target and inhibit protein synthesis. Such activity is in agreement with its robust toxicity observed in all the bacterial hosts tested so far[22,23,30,41]. In contrast with HigB[TAC], no toxicity was detected when HigB2 and HigB3 were expressed in *M. smegmatis* or in *E. coli*. This suggests that both proteins could have a lower affinity for the ribosome or have lost their toxic RNase activity. Although we cannot exclude that these proteins might still be active RNases and toxic when expressed in *M. tuberculosis* in the absence of their endogenous antitoxins, the lack of toxicity of both toxins is supported by the fact that several key active residues of HigB[TAC] (and other closely related RelE-like toxins) are missing in both HigB2 and HigB3, including residues K58, N73, F93 and K99 of the catalytic center, and the newly identified K113 and R117 residues of the long C-terminal helix α3 contacting the initiator tRNA (Supplementary Fig. 10a). In support of this, the structure of HigB[TAC] clearly differs from the alpha-fold models of HigB2 and HigB3 in particular in the two loops between β2-β3 and β4-α3, which contain most of the catalytic residues (Supplementary Fig. 10b). Moreover, the long C-terminal helix α3 of both HigB2 and HigB3, which corresponds to the α3 of HigB[TAC], is negatively charged and may thus not be able to interact with the P-site tRNA (Supplementary Fig. 10c). HigB2 and HigB3 also lack the large positive patch that allows the binding of HigB[TAC] to the 16S rRNA (Supplementary Fig. 10c)

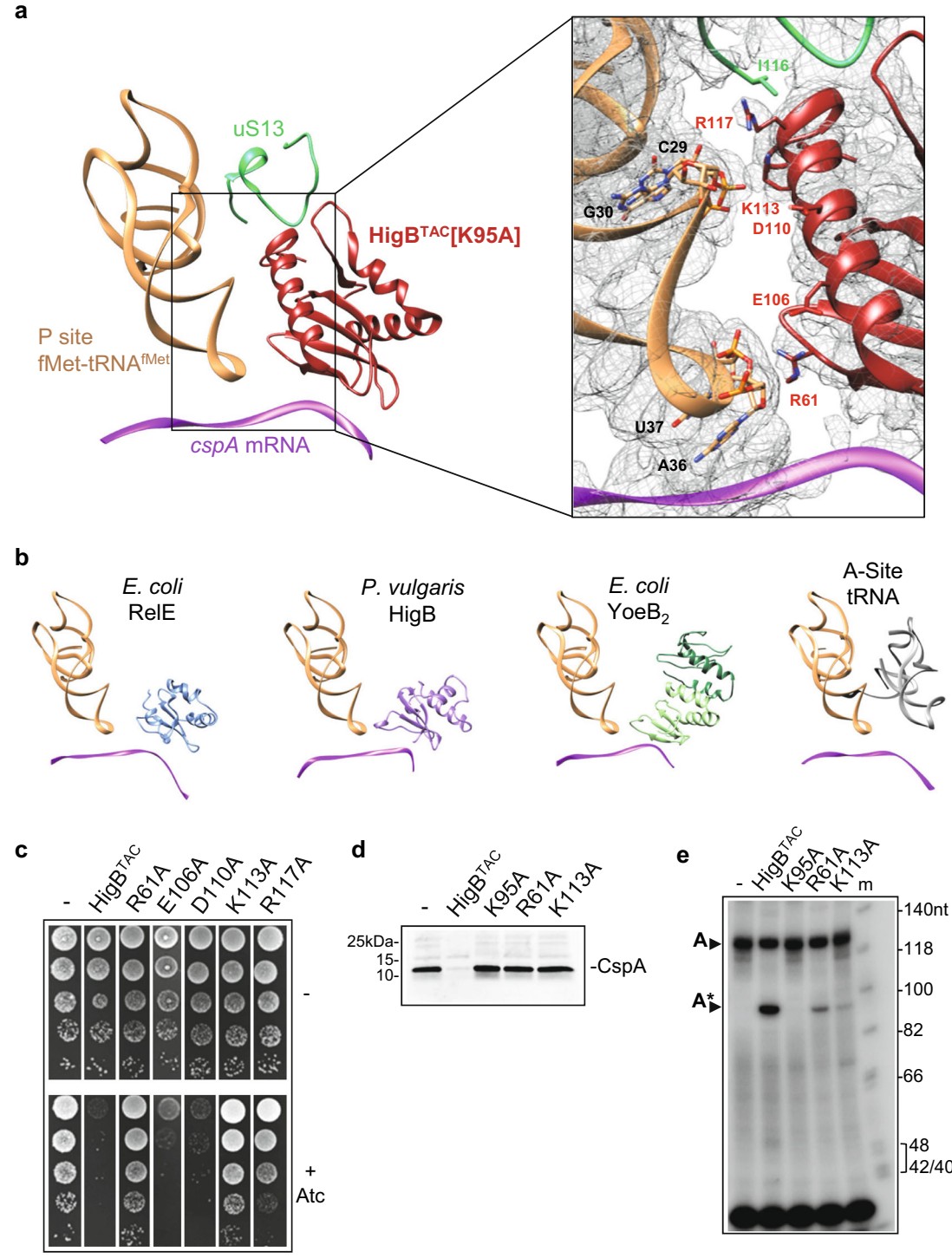

**Fig. 6 The C-terminal extension of the TAC toxin interacts with P-site fMet-tRNA^fMet and is critical for its function. a** Contrary to other ribosome-dependent toxins, HigB^TAC strongly interacts with the P-site tRNA and ribosomal protein S13 (uS13). Left, Overview of the contacts between HigB^TAC [K95A], the P-site fMet-tRNA^fMet, uS13, and the *cspA* mRNA. Right, Close up of the interactions observed between the third α-helix of HigB^TAC [K95A], the P-site fMet-tRNA^fMet, and uS13. Residues and nucleotides within 4 Å of each other are indicated, and the cryo-electron density map is displayed as a gray mesh. **b** Known toxins observed in pre-cleavage state with their respective mRNA targets and the P-site tRNA are shown here with a canonical translation reference. The structures are aligned on the P-site tRNA and presented in the same orientation as in **a**. Shown are (left to right): *E. coli* RelE [R45A-R81A] (PDB 4V7J); *Proteus vulgaris* HigB (PDB 4ZSN); *E. coli* YoeB dimer (PDB 4V8X); and a cognate tRNA observed in the A-site during canonical translation (PDB 7K00). As above, the P-site tRNA is orange and the mRNA is purple. **c** Positively charged residues of HigB^TAC helix α3 in contact with P-site tRNA are important for its function. *M. smegmatis* transformed with pGMC-vector (-), HigB^TAC wild-type or its mutant derivatives (Alanine substitution of residue R61, E106, D110, K113, or R117) were serial diluted, spotted on LB streptomycin agar plates with or without anhydrotetracycline (Atc ng/mL) inducer and incubated 3 days at 37 °C. Alanine substitution in R61 and K113 affects HigB^TAC inhibition of CspA synthesis (**d**) and cleavage (**e**) in vitro. CspA was expressed in a cell-free translation system with or without HigB^TAC wild-type, R61A, or K113A substitution, and analyzed as described in Fig. 3a for CspA protein synthesis and Fig. 3b for *cspA* cleavage. HigB^TAC [K95A] is shown as inactive control. Arrows show the uncleaved (A, 126 nt) and cleaved (A*, 95 nt) *cspA*. Representative results of three independent experiments are shown.

and the hydrophobic pocket shown to host the mRNA is less well defined (Supplementary Fig. 10d). Together these results suggest that HigB2 and HigB3 may neither be able to efficiently bind translating ribosomes nor be active as RNases. Remarkably, a substantial number of *M. tuberculosis* toxins appeared to be non-toxic[4,21–23]. Despite the limited number of conditions tested so far (i.e., mainly expressed in *M. smegmatis* or *E. coli*), these data suggest that selection for less harmful toxin mutants might have occurred in order to limit possible growth disadvantages (or growth inhibition) conferred by an uncontrolled expression of toxins from newly acquired TA genes (or even antitoxin-less toxin genes) via horizontal gene transfer[22,30,54]. In support of this, transposon insertion in at least 10 toxin encoding genes appeared to confer some growth advantages to *M. tuberculosis*[42]. In the case of HigB toxins, the fact that the RelE-like toxin is encoded first in the TA operon could somehow strengthen such a selection for less toxic mutant, as inhibition of newly synthesized toxin by the downstream antitoxin must be well tuned in order to ensure that de novo synthesized toxin is rapidly and efficiently inhibited. Intriguingly, the sole HigB of *M. tuberculosis* that remains active in our conditions is the toxin of the TAC system, in which inhibition of the toxin is performed by the concerted action of the antitoxin and its dedicated chaperone encoded from the same operon[33,34,36].

Activated HigB$^{TAC}$ might randomly scan translating ribosomes until it finds its preferred codon sequences at the ribosomal A-site[29,52]. Our in vivo nEMOTE approach provides a robust identification of the codon sequences recognized by HigB$^{TAC}$, which was further supported in vitro using *cspA* mRNA as a model substrate. It shows that HigB$^{TAC}$ cleaves between the second and the third nucleotide of A-site codons, with the CCA codon encoding a proline being the most represented in vivo (1365 out of the 2037 cleavages identified). CCA is a relatively rare codon in *M. smegmatis* (0.3%) and *M. tuberculosis* (0.6%), but is present in 51 and 75% of their coding sequences, respectively. Although less represented, codons ACA, UCA, UUA or CUA (coding for threonine, serine, leucine, and leucine, respectively) are also well recognized and cleaved by the toxin in vivo and/or in vitro. Overall, HigB$^{TAC}$ appears to have a strong preference for adenosine at the third position, as found for *P. vulgaris* HigB[52,55], with the first and second position also contributing to specificity, although less significantly. Therefore, HigB$^{TAC}$ cleaves irrespectively of the codon family and has the ability to target a large set of mRNAs as long as they are being translated. Despite a robust identification of the preferred motifs, nEMOTE only identified a few annotated mRNA transcripts that are cleaved following HigB$^{TAC}$ expression (18 in total). Four of them encode essential proteins in *M. smegmatis* (seven in *M. tuberculosis*), suggesting that cleavage of such mRNA targets may contribute to the growth inhibition or cell death induced by this toxin. Although the relatively small number of mRNA targets identified could reflect that only a few types of RNA molecules are cleaved, it is more likely that only a fraction of each RNA species is cleaved. In this case, nEMOTE would consistently detect frequent cleavages in abundant RNAs, while cleavages of low-abundance RNAs (and rare cleavages of abundant RNAs) would only be detected in some RNA samples but at the level of background noise in others[44]. An estimation of transcripts abundance in *M. smegmatis* indicates that all the HigB$^{TAC}$ mRNA targets identified in this work have an average, or above average, abundance in vivo when compared to all the annotated transcripts, suggesting that indeed only well-represented transcripts could be identified with confidence (Supplementary Fig. 11). Nevertheless, these data are in agreement with a model in which active HigB$^{TAC}$ would preferentially target highly translated mRNAs, allowing a rapid inhibition of protein synthesis[49]. Whether such activity would

provide a pool of free ribosomes that would ensure bacterial growth once normal conditions resume is not known.

Despite similar fold, ribosome-dependent RelE-like toxins are poorly related in sequences, even for their catalytic site[29,49,51,53], and exhibit important differences in codon preference. Indeed, *V. cholerae* HigB1 was shown to preferentially cleave AAA, ACU, GCG, and GCA, whereas AAA, ACU, AAU, CUG, GUG and GCG are cleaved in vivo by *Vc*HigB2[56]. Other studies revealed that *P. vulgaris* HigB cleaves AAA in vivo[55] and adenosine rich codons in vitro[52], *E. coli* YoeB cleaves AUG, UAA, CUG, GCG and GCU in vitro[53] and YafQ cleaves AAA in vivo[57]. The *E. coli* RelE, which exhibits little specificity in vivo, was shown to preferentially cleaves codons with purines at positions 2 and 3 of the A-site codon[29,58]. Finally, a recent work performed in vivo in *E. coli* with multiple RNase toxins showed that both RelE and HigB favor guanosine after the cleavage site and YoeB adenosine before the cleavage site within codons[59]. Although the conditions and organisms used to test toxin activities were generally different (comparison should thus be taken with caution), these data suggest relatively few overlaps with HigB$^{TAC}$, especially in vivo where none of the four most represented codons targeted by HigB$^{TAC}$ were identified as preferred targets for the other toxins (except for RelE that shows little specificity in vivo). Besides, the AAA codon, which is efficiently recognized by most of the toxins described above was not identified in our in vivo work and was only poorly cleaved by HigB$^{TAC}$ in vitro when compared to CCA or other preferred codons (Supplementary Fig. 3c). This further highlights the remarkable reservoir of substrate preferences found among RelE-like toxins so far.

We found that 3 out of the 10 annotated transcripts of *M. smegmatis* that are recognized by HigB$^{TAC}$ are cleaved at a CCA codon located at position 2, just after the initiation codon (i.e., *cspA*, *rpsA*, and *rpsL*). Together these 3 transcripts account for about 37% (761/2037) of all the cleavage events detected in vivo. Yet, while more than half of all the annotated coding sequences in *M. smegmatis* or *M. tuberculosis* contain at least one CCA (see above), only 0.84 and 1.4% have a CCA at this position, respectively. These data indicate that despite the fact that cleavage can occur all along a targeted mRNA, there is a preference for HigB$^{TAC}$ to cleave a CCA at the second position, both in vivo and in vitro. Translation initiation could thus offer a window of opportunity for the toxin to reach the A-site, as previously proposed[50]. Indeed, a comparison of the ribosome-bound HigB$^{TAC}$ with the structure of the ribosomal initiation complexes (IC) shows that HigB$^{TAC}$ cannot bind the 30S-IC when IF1 is present[60]. However, HigB$^{TAC}$ could easily accommodate into the late 70S-IC, even if IF2 is still present (Supplementary Fig. 12). Noticeably, crucial contact between the long C-terminal helix α3 of HigB$^{TAC}$ and the P-site fMet-tRNA described in this work would still be possible, even if the fMet-tRNA is in P/I conformation. The fact that initiation is by far the slowest step of translation[61,62] and that the presence of IF2 prevents tRNAs from accessing the A-site and competing with the toxin, suggests that HigB$^{TAC}$ would have more chances to recognize and cleave at the CCA adjacent to the initiation codon than at other codons further down the sequence. Such a mechanism would contribute to the efficient and rapid inhibition of protein synthesis and the resulting growth inhibition caused by HigB$^{TAC}$. Whether the specific contacts found between the HigB$^{TAC}$ C-terminal extension and the fMet-tRNA facilitate such a process remains unknown. Further work is warranted to address such a possible scenario.

## Methods
**Bacterial strains and culture conditions**. *E. coli* strains W3110[34], MG1655(ATCC 700926), BL21(λDE3) (Novagen), BL21 AI(λDE3) (Novagen), *M. smegmatis* mc²155 (strain ATCC 700084) and mc²155 *rnj102* (Δ*rnj*)[46] were previously

described. DNA cloning experiments were carried out in *E. coli* DH5α (NEB) or Stellar (Clontech). *E. coli* strains were grown in Luria Bertani medium (LB) and supplemented with streptomycin (25 μg/mL), ampicillin (50 μg/mL), or kanamycin (50 μg/mL) as required. *M. smegmatis* strains were grown in LB medium supplemented when necessary with streptomycin (25 μg/mL) or kanamycin (50 μg/mL), and Tween80 (0.05%) to minimize cell aggregation in liquid culture.

**Western blot analysis**. Whole-cell extracts were performed as described[35]. Briefly, 1 ml aliquots of the cell cultures were collected at 5000 x g and resuspended in 1× SDS loading buffer (1/4th volume of the initial OD$_{600}$). Western blots monitoring GFP or Strep-tagged GroES in vitro synthesis using the PURE system with or without HigB$^{TAC}$ toxin were performed by loading 10 μL of PURE reaction with 1× SDS loading buffer. Proteins were then separated on Mini-Protein TGX gels (Bio-Rad) by SDS-PAGE and transferred to polyvinylidene difluoride membranes (Bio-Rad) using the Trans-Blot® TurboTM transfer system (Bio-Rad). Membranes were blocked for 1 h at room temperature in 5% (w/v) nonfat dry milk in PBS containing 0.05% (v/v) Tween 20. Primary antibodies used in this study were anti-HigB$^{TAC}$ antibody (dilution 1:1000)[30], anti-GFP (Invitrogen, 1:3000), StrepMAB-Classic horseradish peroxidase (HRP; Iba Life Sciences, 1:30,000), and anti-His6-HRP (Invitrogen, 1:1000). HRP-conjugated mouse IgG (Promega, 1:2500) was used as a secondary antibody. Blots were developed by chemiluminescence using Clarity Western ECL substrate (Bio-Rad) with the ChemidocTM Touch imaging system (Bio-Rad) and analyzed with the Image Lab software (Bio-Rad).

**RNA isolation for the nEMOTE procedure**. Total RNA was isolated from *M. smegmatis* strains WT/ΔrnJ that were transformed by electroporation with pGMC-toxins and grown to mid-exponential phase. Independent cultures of each strain were launched on different days and with different batches of the medium. When 10 mL cultures reached an OD$_{600}$ of 0.4, anhydrotetracycline was added to a final concentration of 100 ng/mL to induce toxins for 3 or 24 h. After induction, 1 mL of culture was serial diluted on a LB agar plate supplemented with streptomycin and with or without ATc to check the activity of the toxins and the presence of potential suppressor before RNA extraction. Plates were stored at 37 °C for 3 days. Cells were pelleted by centrifugation at 2800 × g for 10 min at 4 °C and the supernatants were removed. Then, 1 mL of cold ethanol/acetone (1:1, v/v) was immediately added to the pellets to protect the RNA (note that these pellets could be kept at −80 °C). Next, the mix ethanol/acetone was removed and a mix of 700 μL of buffer RLT (RNeasy® Mini Kit. QIAGEN) + 10 μL β-mercaptoethanol was added to resuspend the pellet. For each sample, 25–50 mg acid-washed glass beads (500 μm diameter Sigma Aldrich) were weighed in a 2 mL Safe-Lock tube. The cells were broken using a BeadBeater (Bertin Technologies Precellys 24) 1 min ON–1 min OFF–1 min ON. The lysed cells were centrifuged at 15,800 × g at 4 °C for 10 min. The supernatants were transferred in a new RNase-free Eppendorf tubes. Then, 70% Ethanol v/v is added to the supernatants, mixed, and transferred to the provided column (RNeasy® Mini Kit. QIAGEN). Finally, purification of total RNA using the RNeasy® Mini Kit was completed according to the manufacturer's guidelines. RNA quality was verified with an Agilent 2100 BioAnalyzer (Genotoul platform in Toulouse).

**nEMOTE library preparation**. The protocol for nEMOTE was carried out as described[43], with slightly modified primers (Supplementary Table 3). Briefly, 8 μg of RNA were incubated with XRN-1 (NEB) for 4 h to digest mono-phosphorylated RNA. Then the RNA was split into two pools, one pool of RNA was both phosphorylated with polynucleotide kinase (NEB) and ligated to Rp8 oligo, the control pool of RNA was only ligated to Rp8. The Rp8 oligo that we ligate to the 5'OH ends of the RNA contains a unique molecular identifier (a series of random nucleotides), which allows us to tag each ligation event with its own unique barcode. Reverse transcription was performed using DRNA primer and ProtoScript II RT enzyme (NEB). Finally, PCR amplification was performed with barcode primers and Illumina adaptator A and B primer using Q5 Polymerase Hot-Start (NEB).

**Bioinformatics analyses**. Raw sequencing reads were first filtered and trimmed using the emoteStepI method from a Perl program called EMOTE-conv[63]. The trimmed reads are mapped against the *M. smegmatis* genome (NC_008596.1) with bowtie (version 1.2)[64], using the parameters -a --best –strata -v 1. The obtained alignment file is submitted to the emoteStepII step from EMOTE-conv in order to compute an annotated coverage table corresponding to the number of reads per position per EMOTE barcode. The unique molecular identifier that was included in the Rp8 oligo allows us to detect when the cDNA from a single ligated RNA molecule is represented by one or more different illumina reads (this is possible due to the PCR amplification in the library preparations). Thus, all identical reads with the same unique molecular identifier are only counted as a single RNA cleavage event. Downstream analysis and plots are performed in R (version 3.4.4, running under Ubuntu 14.04.6). We set a very conservative cut-off for what was considered a true toxin-dependent cleavage event: The genomic positions where the 5' OH-ends of at least 7 independent RNA molecules were detected for at least one test sample (toxin + PNK) and where no 5'OH-ends were detected in all three negative control samples (toxin-PNK, vector+PNK, and vector-PNK). The putative cut-site motif is plotted with the R package ggseqlogo (version 0.1).

The evolutionary history between *M. tuberculosis* HigB$^{TAC}$, HigB2, HigB3, and *E. coli* RelE was inferred using the maximum likelihood method. Protein sequences were aligned by PROMALS3D and the tree was generated by MEGA11. The tree with the highest log likelihood (−1228.63) is presented in Fig. 1b. The tree is drawn to scale, with branch length representing the number of substitutions per site.

**M. smegmatis gene expression**. To generate the violin plot showing *M. smegmatis* gene expression in transcripts per million the following procedure was applied. The data from *M. smegmatis* wild-type mRNA sequenced with Illumina[65] consists of 2 replicates: Replicate 1 (112309244 SE reads of 51 bp) and Replicate 2 (126869005 SE reads of 51 bp). After Trimmomatic cleaning, the cleaned reads, replicate 1 (24637386 SE reads), replicate 2 (67623219 SE reads), were remapped to the reference genome of *M. smegmatis* MC$^2$155 (CP000480) with bwa and converted to sorted bam with Samtools. Since no differential analysis was possible with two replicates of the same condition, we decided to calculate the TPM (transcripts per million) for all genes of both replicates and compare them by their mean values. The TPM and read counts were obtained from the bam and TPMCalculator[66].

**Cell-free transcription–translation system in vitro**. Cell-free transcription/translation in vitro assay using the PURE system (NEB) was carried out as described[34]. Briefly, DNA of cspA (Rv3648c) 204 bp, gfp 717 bp, or groES (Rv3418c) 303 bp were amplified by PCR using primers containing T7 promoter and terminator and added at a final concentration of 20 ng/μL to the PURE system with or without toxins (1.5 μM). Protein synthesis was performed at 37 °C for 1 h 30 min in the presence of 0.6 μCi/μL of $^{35}$S-methionine. Samples were then separated by SDS/PAGE on 4–20% Mini-Protean TGX gels (Bio-Rad) for 1 h 15 min at 100 V. Gels were fixed in 10% acetic acid/40% methanol (v/v) for 30 min and proteins were visualized using a Typhoon phosphorimager (GE Healthcare) and Multigauge software (Fuji).

**Primer extension**. For cspA RNA extracts without ribosome, DNA fragment of cspA containing T7 promoter and terminator (forward primers CspA_CCA_Fw, CspA_CCT_Fw/CspA_CCG_Fw/CspA_CCC_Fw, CspA_CAA_Fw/CspA_CTA_Fw/ CspA_CGA_Fw, CspA_ACA_Fw/CspA_TCA_Fw/CspA_GCA_Fw, CspA_codon 7_Fw, OOF1_Fw, OOF2_Fw, or CspA_AAA_Fw, and the reverse primer CspA_term_Rv listed in Supplementary Table 3), as prepared for the cell-free transcription–translation system in vitro, was transcribed using 40 unit of T7 RNA polymerase (Promega) with or without HigB$^{TAC}$ (1.5 μM) for 2 h at 37 °C. RNA was subsequently extracted with TRI Reagent (MRC) and chloroform. For cspA RNA extracts with the ribosome, the DNA fragment of cspA containing T7 promoter and terminator was added to the cell-free transcription–translation system as described above, except that $^{35}$S-Methionine was not included. After 2 h at 37 °C, RNA was extracted with TRI Reagent (MRC) and chloroform. For the reverse transcription, up to 2 μg of purified cspA RNA, 0.05 μM $^{32}$P-labeled cspA primer (primer RT CspA P32 PAGE 1, except in Supplementary Fig. 4b; Supplementary Table 3 where primer RT CspA P32 PAGE 2 was used in order to obtain longer cspA fragments that included the P17 and P22 region) and 1 mM dNTPs were mixed in a 10 μL volume, incubated at 65 °C for 5 min, and chilled on ice for 1 min. Finally, 10 μL 2× buffer (mix 4 μL 5× ProtoScript II RT (NEB), 2 μL 0.1 M DTT, 8 units RNasin® Plus Ribonuclease Inhibitor (Promega), 200 units of ProtoScript II RT Enzyme (NEB) in 10 μL) were mixed and incubated at 48 °C for 1 h. cDNA was mixed with RNA loading dye, loaded on a 6% Polyacrylamide gel containing 7 M urea, separated at 300 V for 1 h 15 min, and revealed by autoradiography using Typhoon phosphorimager (GE Healthcare) and Multigauge (Fuji Film). Note that $^{32}$P labeling of the primer was performed using PNK (10 u, Thermofisher) at 37 °C for 1 h in the presence of the primer RT CspA P32 PAGE 1 or 2 (0.5 μM final concentration) and 2,5μCi/μL of $^{32}$P. The labeled primer was purified with Bio-Spin® 6 Columns (Biorad).

**Protein purification**. Purification of HigB$^{TAC}$, its mutant derivatives, HigB2, and HigB3 with a C-terminal His$_6$ tag was performed as follows. About 50 fresh colonies of *E. coli* BL21 AI transformed with pET20b-toxin derivatives were pooled and grown in 1 liter (or 250 mL for HigB$^{TAC}$ inactive mutants) of LB ampicillin supplemented with glucose (0.2% w/v) at 37 °C until OD$_{600}$ reaches 0.5-0.7. Cell cultures were centrifuged at 6000 × g for 10 min, resuspended in LB ampicillin supplemented with 0.2% of L-arabinose to induce toxin expression, and incubated overnight at 22 °C. Cells were centrifuged and lysis was performed by resuspending the pellets in 20 mL of lysis buffer (25 mM equimolar solution of Na$_2$HPO$_4$/NaH$_2$PO$_4$; 200 mM NaCl; 20 mM imidazole pH 8.0) supplemented with one EDTA-free Protease Inhibitor tablets (Roche) and benzonase 500 unit (Sigma-Aldrich) followed by lysis in a cell disruptor at 1.5 bar (One shot model, constant systems Ltd). Intact cells were removed by centrifugation at 30,000 × g for 30 min at 4 °C. After washing the Ni-NTA column 3 times with the wash buffer, the supernatant was applied to the column for 30 min. Finally, the protein was eluted with Elution buffer (25 mM equimolar solution of Na$_2$HPO$_4$/NaH$_2$PO$_4$; 200 mM NaCl; 250 mM imidazole, pH 8.0). Proteins were dialyzed with the exchange buffer (25 mM equimolar solution of Na$_2$HPO$_4$/NaH$_2$PO$_4$; 200 mM NaCl; 10% (w/v) glycerol; pH 8.0) using PD-10 MiniTrap (GE-life science) and concentrated if

needed with Vivaspin (5 kDa pore size. Sartorius) and stock at −80 °C. For X-ray crystallography analysis, fresh colonies of pET15b-*Mtb*-HigB$^{TAC}$[K95A] transformed in *E. coli* BL21 (DE3) were pooled in 20 mL LB broth supplemented with ampicillin (100 µg/mL) and grown overnight at 37 °C. Cultures were carried out by inoculating 1 L of LB broth (+ampicillin 100 µg/mL) with 15 mL of pre-culture at 37 °C at 180 rpm. Heterologous protein expression was induced by adding 1 mM IPTG when OD$_{600}$ reached ~0.5. Cells were allowed to grow for 4 h at 37 °C. The pelleted cells were suspended in 80 mL of buffer A (25 mM Na$_2$HPO$_4$/NaH$_2$PO$_4$, 200 mM NaCl, pH 7.0) supplemented with 20 mM imidazole, 0.2 mM PMSF, 0.5 mg/mL lysozyme and lysed using 4 cycles of cell disruptor (EmulsiFlex-C5, Avestin) at about 10,000 psi prior to centrifugation at 25,000 × g for 30 min at 4 °C. The supernatant was applied to a 1-mL HisTrap$^{TM}$ HP column (Cytiva) equilibrated with buffer A containing 20 mM imidazole. After extensive washing with 30 mM and 60 mM imidazole in buffer A, the recombinant protein was eluted with buffer A supplemented with 150 mM imidazole. The eluted protein was dialyzed twice against 1 L of buffer A without imidazole under magnetic stirring at 4 °C and the His$_6$-tag was then cleaved with 0.1 U of thrombin per mg of protein in the presence of 10X cleavage buffer (200 mM Tris-HCl, 1.5 M NaCl, 25 mM CaCl$_2$, pH 8.4) overnight at 12 °C under rotary agitation. The cleaved protein was injected into a 1-mL HisTrap$^{TM}$ HP column (Cytiva) equilibrated with buffer A supplemented with 20 mM imidazole and recovered from the flow-through, then concentrated using a Vivaspin 5 centrifugal concentrator (Sartorius) to about 7 mg/mL, prior to injection into a HiLoad 16/60 Superdex 75 (GE Healthcare) pre-equilibrated with 20 mM MES, 200 mM, NaCl pH 6.5 (buffer B). MenT3$^{K189}$ was purified as described[4] and Bovine Serum Albumin (BSA) was purchased from Sigma (ref.A3059).

**Crystallization and crystal structure determination.** Purified HigB$^{TAC}$[K95A] was concentrated to 28.6 mg/mL in buffer B and crystallized at 12 °C in 30% (w/w) PEG 4,000, 0.2 M sodium acetate, 0.1 M sodium acetate, pH 4.6. The volume ratio between the purified toxin and crystallization solutions was 200:100 (nL). Crystals were directly flash-frozen in liquid nitrogen. Datasets were collected on beamline MASSIF-3 (ID30-A3) at the European Synchrotron Radiation Facility (ESRF, Grenoble, France). All data were indexed, integrated, and scaled using XDS version Jan 31, 2020, and the CCP4 v7.0.078 software suite was used for subsequent crystallographic calculations[67]. The structure was solved by molecular replacement using PHASER v2.8.3[68] and a structure-based homology model derived from the structure of *S. pneumoniae* HigB (PDB 6AF4). The search model was truncated to a polyalanine and all loops were removed to avoid model bias. Iterative cycles of manual model building in COOT v0.9.6[69] and refinement procedures using REFMAC v5.8.0258[70] were applied until convergence. Details of data collection, cell parameters, processing, and refinement statistics are presented in Supplementary Table 1.

**Sample preparation for cryo-EM.** For the cryo-EM complex, ribosomes were purified from *E. coli* MG1655. When the culture reached an OD$_{600}$ of 0.8, cells were pelleted, resuspended in FP buffer (20 mM Tris-HCl pH 7.5, 50 mM MgOAc, 100 mM NH$_4$Cl, 0.5 mM EDTA, and 1 mM DTT) and lysed in a French press. The lysate was then clarified by centrifugation at 20,000 × g for 45 min at 4 °C. Next, the supernatant was layered 1:1 (v/v) over a high-salt sucrose cushion buffer (10 mM Tris-HCl pH 7.5, 10 mM MgOAc, 500 mM NH$_4$Cl, 0.5 mM EDTA, 1.1 M sucrose and 1 mM DTT). After ultracentrifugation at 92,000 × g for 20 h at 4 °C, the resulting ribosome pellets were resuspended in 1 mL of 'Ribo_A' buffer (10 mM Tris-HCl pH 7.5, 10 mM MgCl$_2$, 50 mM NH$_4$Cl, 0.5 mM EDTA and 1 mM DTT). To isolate the 70 S ribosomes from 30 S and 50 S ribosomal subunits, the ribosomes were centrifuged at 95,000 × g for 18 h at 4 °C through a 10–45% (w/w) linear sucrose gradient in Ribo_A buffer. Gradients were fractionated before determining the A$_{260}$ absorbance profiles. Fractions corresponding to the 70S peak were mixed and diluted in Ribo_A buffer for final ultracentrifugation at 92,000 × g for 20 h at 4 °C. The ribosomal pellets were resuspended in Ribo_A buffer, and flash frozen and stored at −80 °C.

To prepare the complex, 25 pmol of fMet-tRNA$^{fMet}$ (VWR, Ref. ICNA0219915410) was first refolded for 2 min at 80 °C in "Buffer I" (10 mM HEPES-KOH pH 7.5, 25 mM MgCl$_2$, 25 mM, and 20 mM NH$_4$Cl), and this was followed by a second incubation at room temperature for 30 min. Next, purified 70 S ribosomes (12.5 pmol) were incubated at 37 °C for 15 min in "Buffer-III" (10 mM MgOAc, 10 mM NH$_4$Cl, 50 mM KCl, 5 mM HEPES-KOH pH 7.5, and 1 mM DTT) with 25 pmol of *cspA* mRNA and 25 pmol of the folded fMet-tRNA$^{fMet}$. Finally, 1250 pmol of HigB$^{TAC}$ [K95A] toxin were added, and this was incubated at 37 °C for 5 min. After adjusting concentrations to 160 nM in buffer-III, samples were directly applied to glow-discharge holey carbon films (Quantifoil 3.5/1 µm). These grids were flash-frozen in liquid ethane using a Vitrobot Mark III (FEI).

**Cryo-EM data collection.** The frozen grids were then transferred to the Structural Biophysical Chemistry Platform of the IECB, where they were imaged using a Talos Arctica cryo-TEM (Thermo Fisher Scientific) operating at 200 kV and equipped with a field-emission gun. SerialEM v3.8 software was used to automatically record 6381 movies under low-dose conditions on a K2 direct electron detector (Gatan) with a defocus range of 0.4–2.0 µm and at a final pixel size of 0.9291 Å.

**Image processing.** Movies were corrected for the effects of drift and beam-induced motion using MotionCor2 v1.0.6 software[71]. Contrast transfer function (CTF) parameters were estimated using Gctf v1.18 software[72]. Electron micrographs showing signs of drift or astigmatism were discarded, resulting in a dataset of 5178 movies. Particles were semi-automatically selected in Cryosparc v2.12[73] and subjected to two rounds of 2D classification in order to remove defective particles. This resulted in the selection of 284,820 particles. All subsequent data processing was performed using RELION v3.1.3[74]. An initial three-dimensional (3D) auto-refinement using a large soft circular mask (diameter 380 Å) produced a reconstruction at a resolution of 3.33 Å. The pixel size was then re-estimated by comparison to the atomic model of the *E. coli* mature 70S subunit (PDB ID 4YBB), and adjusted to 0.893 Å/pixel. To improve the homogeneity, the datasets were then sorted into 12 subsets using the 3D-classification function. This resulted in the following grouping: 70S ribosomes (7 classes containing 175,859 particles); ratcheted ribosomes (2 classes and 39,752 particles); the 50S ribosomal subunit (2 classes and 51,812 particles); and poorly resolved particles (1 class of 27,397 particles). The particles which were clearly homogenous 70S or ratcheted ribosomes were further processed separately using the same protocol. This was followed by a second round of 3D auto-refinement using the same parameters, resulting in reconstructions with resolutions of 3.4 Å (for the 70S) and 4.21 Å (for the ratcheted ribosomes). We then subtracted the ribosome signal from the datasets by using a soft mask generated from the previous refinement run (voxel values of 0 inside, 1 outside, extended by 6 pixels, and a soft edge of 6 pixels). The subtracted datasets were then sorted by 3D classification without alignment, using a tight soft spherical mask with a diameter of 200 Å. We tested various combinations in order to split the datasets into as many 3D classes as possible while still keeping the groups homogenous; however, the HigB$^{TAC}$[K95A] toxin could not be found in the dataset with the ratcheted ribosomes. The dataset containing the 70S ribosomes was separated into eight classes, with one containing HigB$^{TAC}$[K95A]. The 42,261 particles corresponding to that class were then selected and reverted to their original content, before the subtraction of the ribosome signal. A third round of 3D auto-refinement resulted in a 3.8-Å map of the ribosome containing the CspA mRNA, an fMet-tRNA$^{fMet}$, and HigB$^{TAC}$[K95A]. To ensure that a homogeneous class was indeed obtained, 3D classification with signal subtraction was again performed, this time using a soft mask with a diameter of 80 Å to focus on the toxin. This dataset was split into four classes, with one class of 13,877 particles containing the toxin. Once again, we reverted to the original non-subtracted particles and reconstructed a new map at a resolution of 4.85 Å. After CTF refinement and particle polishing, the 'shiny' particles were subjected to a final round of 3D auto-refinement and post-processing. This resulted in a consensus map with an overall resolution of 3.11 Å as per RELION's gold-standard Fourier shell correlation (FSC) calculation[75]. To take into account the internal flexibility of the ribosome, we also performed a multibody refinement[76]. The ribosome was separated into the large 50S subunit, the body of the small 30S subunit, and the 30S head. HigB$^{TAC}$[K95A] and CspA mRNA were only included in the two 30 S fragments, while the tRNA was included in all three sections. Masks corresponding to these sections were made from a 30-Å low-pass filtered version of the consensus map, with soft edges of 12 Å in order to define the solvent region boundary and to ensure that all the sections overlapped. The multibody maps were then fitted and resampled on the consensus map using UCSF-Chimera v1.13.1[77]. Finally, the four maps were sharpened using Phenix v1.18.2[78], their local resolutions estimated using Resmap v1.1.4, and map quality was assessed using Phenix mtriage[79].

**Model building and refinement.** UCSF-Chimera v1.13.1[77] was used to rigid-body fit the cryo-EM structure of the *E. coli* ribosome at a resolution of 2 Å (PDB 7K00) into our maps, with each protein and RNA treated separately. Our crystal structure of *M. tuberculosis* HigB$^{TAC}$[K95A] toxin (PDB code 7AWK) was then fitted in the remaining portion of the map. The different molecules were manually adjusted in their respective multi-body maps using COOT v0.9.5. Special attention was paid to the toxin, the tRNAfMet, and the CspA mRNA, and their models were compared with the different multibody maps. The final atomic model was further improved by real-space refinement against the consensus maps using Phenix v1.18.2. Outliers were then again manually corrected in COOT and the model was refined a second time in Phenix. Once the structure was complete, Mg2+ ions were manually added in COOT using the "unmodelled blobs" function and a threshold of 5.5 RMSD. The model quality was evaluated with MolProbity v4.5.1[80]. The remaining analysis and the illustrations were done using UCSF-Chimera v1.13.1[77]. Details of data collection, processing, and refinement statistics are presented in Supplementary Table 2, and a schematic representation of the cryo-EM single-particle reconstruction workflow is in Supplementary Fig. 13. Note that full scan blots and source data can be found in the Source Data file and in the Supplementary Information file.

**Reporting summary.** Further information on research design is available in the Nature Research Reporting Summary linked to this article.

## Data availability
The data that support this study are available from the corresponding author upon reasonable request. The electron density maps and structure models are deposited in the

EMDB and PDB under the following accession codes, respectively: 7AWK for the crystal structure of the *M. Tuberculosis* HigB^TAC [K95A] toxin alone; and EMD-12261 and 7NBU for the toxin and its target mRNA on the translating *E. coli* ribosome. The nEMOTE sequencing data have been deposited in Zenodo [https://doi.org/10.5281/zenodo.6397033]. Source data are provided with this paper.

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

## Acknowledgements

We thank Pauline Texier et Michele Coddeville for plasmid gifts and for their assistance during the course of this study, and Patrick Viollier for useful advice. We also thank Israel Mares-Mejía for his help with toxin purification. This work has benefited from the facilities of the Microscopy Rennes Imaging Center (Mric) and from the facilities and expertise of the Structural Biophysico-Chemistry platform (BPCS) at the IECB (CNRS UMS3033, Inserm US001, Bordeaux University). We thank the scientific staff of the European Synchrotron Radiation Facility (Grenoble, France). We thank the staff of beamline MASSIF-3 (ID30-A3) at the European Synchrotron Radiation Facility where the crystallographic experiments were conducted. The crystallization and macro-molecular crystallography equipment used in this study are part of the Integrated Screening Platform of Toulouse (PICT, IPBS, IBiSA). We also thank the iGE3 genomics platform at the University of Geneva. This work was supported by Agence Nationale de la Recherche (ANR-19-CE12-0026) to P.G., R.G., and L.M, Programme d'Investisse-ments d'Avenir (ANR-20-PAMR-0005) to P.G., by the National Natural Science Foun-dation of China (32000021) to X.X., by Fondation pour la Recherche Médicale (FDT201805005796) to M.M., and by the Swiss National Science Foundation (CRSII3_160703) to P.G. and L.F.

## Author contributions

Performed research: M.M., E.G., X.X., P.B., V.G., D.-J.B., N.S., S.C.; analyzed data: M.M., E.G., X.X., H.A., P.B., V.G., D.-J.B., N.S., G.D., S.C., P.R., L.F., L.M., R.G., P.G.; designed research: M.M., E.G., X.X., P.B., V.G., D.-J.B., N.S., S.C., P.R., L.F., L.M., R.G., P.G.; wrote the paper: M.M., E.G., and P.G. with contributions from all the authors. R.G. and P.G. supervised the study.

## Competing interests

The authors declare no competing interests.
