## [Peer Review File · Nature Communications]

Substrate recognition and cryo-EM structure of the ribosome-bound TAC toxin of *Mycobacterium tuberculosis*Editorial Note: This manuscript has been previously reviewed at another journal that is not operating a transparent peer review scheme. This document only contains reviewer comments and rebuttal letters for versions considered at *Nature Communications*.

REVIEWERS' COMMENTS

Reviewer #1 (Remarks to the Author):

In this revision, the authors have addressed all comments adequately. The manuscript now presents a comprehensive study and a competent discussion that presents the data in the context of the field.

Two small typos could be fixed:

1) The usage of figure numbers for supplementary figures is inconsistent. (for example, l 110 Fig. S1a but line 114 Supplementary Fig. 1c). Either use S1, S2 or Supplementary 1, 2, etc.

2) Suppl Materials p. 4, last sentence: 'sequenced verified' is awkward. Please rephrase the sentence.

Reviewer #2 (Remarks to the Author):

The authors have addressed my concerns in a satisfactory and very thorough way, so I have no further points to make.

Reviewer #4 (Remarks to the Author):

The paper by Mansour et al., explores the function and regulation of several HigBA TA modules from *M. tuberculosis* with particular focus on HigBA-TAC. They provided evidence that HigBA-TAC is a ribonuclease that cleaves mRNA in a ribosome and codon specific manner. They also determined the cryo-EM structure of the ribosome-bound TAC HigB revealing the specific mechanism by which the TAC toxin interacts with the ribosome and the P-site tRNA. They also characterised the toxicity of other higBA (higBA2 and 3) and showed that these TA modules have only limited toxicity and proposed that this reduced toxicity may provide fitness to *Mtb* compared to systems with a more binary output.

Overall the manuscript brings forth consistent and sound data and shows a new way in which HigB/RelE toxins can be anchored at the A-site. In my opinion, the findings will be of interests for the readers of Nat Comm and could be published after some revisions.

Major point:

For me the description of HigB2 and HigB3 as “inactive toxins” remains a thorny issue that is still not covered convincingly because proper biochemistry was not done. The authors mentioned that the drop in GFP synthesis could be due to interference with translation which is true. However these HigB toxins are all ribosome-dependent, without ribosomes they are all inactive, HigB2 and 3 could just “look” inactive because they have lower affinity for ribosomes but once loaded, may be as active as HigB-TAC. Could the authors comment on this?

Minor comments:

- Page 3, the authors mentioned that the functions of TA modules remain largely unknown, but then seemingly contradict themselves. This type of sentences have dominated the narrative in papers involving TA modules as a way to avoid recent issues regarding TA functions. While a unique overarching function has been strongly challenged, the function of TAs in plasmid maintenance and more recently in bacterial defence as abortive systems or in phages is well documented (particularly relevant for mycobacterium and demonstrated by Dedrik et al. Nature Microbiology 2017). So this part should be amended accordingly.

The main part and methods section lacks any description about the purification of ribosomes or details on how the ribosome-HigB complex was assembled and processed to be used in the grids (only that HigB was used in “high excess” which is completely meaningless). This contrast with the very detailed section on image processing and a major oversight (the authors don’t even mentioned the source of ribosomes, which is very relevant in this case). This should be explained in details in the final version of the paper otherwise it would be very difficult to repeat this structure or this protocol for a similar toxin.

It was not justified or elaborated why 7-fold was considered as threshold to chose a cleavage site.

In general the authors tend to abuse terms such “significant, high, low, likely”. They should be precise.

When comparing structures, please provide structural alignments, otherwise interpreting results becomes very difficult, specially when discussing substitutions between different TA toxins (for example in Fig 6b). Without structural alignments is for example very hard to interpret and follow why HigB2 and HigB3 are not toxic based on the the relevant residues and substitutions in HigB-TAC. This point was already raised by other reviewers and the current Fig are still not great in this regard.

The authors also used unnecessarily the overlay of electron densities in the figures. While it is good they showed it perhaps once like in Fig. 4 to show that HigB is indeed bound, in the rest of the figures that convey mechanistic information, this only difficult the interpretation and should be removed or shown as supplementary, keeping the main figures without the densities (this is critical in Fig 5).

Response to Editor and Referees

We would like to thank you and Reviewers for their positive comments and constructive criticism. Point-by-point responses to Reviewer comments are listed below in blue, and where necessary, changes have been made to the original manuscript and are shown in yellow highlight. In some instances, changes have been made to figures but these are not altered in color, in order to maintain consistency within the figure.

We thank the reviewers for their positive evaluation of our work.

REVIEWERS' COMMENTS

Reviewer #1 (Remarks to the Author):

In this revision, the authors have addressed all comments adequately. The manuscript now presents a comprehensive study and a competent discussion that presents the data in the context of the field.

Two small typos could be fixed:

1) The usage of figure numbers for supplementary figures is inconsistent. (for example, l 110 Fig. S1a but line 114 Supplementary Fig. 1c). Either use S1, S2 or Supplementary 1, 2, etc.

Correction of numbers for supplementary figures have been made throughout the revised manuscript.

2) Suppl Materials p. 4, last sentence: 'sequenced verified' is awkward. Please rephrase the sentence.

The sentence from the "Supplementary Methods" section was corrected for: "All the plasmids were sequence-verified."

Reviewer #2 (Remarks to the Author):

The authors have addressed my concerns in a satisfactory and very thorough way, so I have no further points to make.

Reviewer #4 (Remarks to the Author):

The paper by Mansour et al., explores the function and regulation of several HigBA TA modules from *M. tuberculosis* with particular focus on HigBA-TAC. They provided evidence that HigBA-TAC is a ribonuclease that cleaves mRNA in a ribosome and codon specific manner. They also determined the cryo-EM structure of the ribosome-bound TAC HigB revealing the specific mechanism by which the TAC toxin interacts with the ribosome and the P-site tRNA. They also characterised the toxicity of other higBA (higBA2 and 3) and showed that these TA modules have only limited toxicity and proposed that this reduced toxicity may provide fitness to *Mtb* compared to systems with a more binary output. Overall the manuscript brings forth consistent and sound data and shows a new way in which HigB/RelE

toxins can be anchored at the A-site. In my opinion, the findings will be of interests for the readers of Nat Comm and could be published after some revisions.

Major point:

For me the description of HigB2 and HigB3 as “inactive toxins” remains a thorny issue that is still not covered convincingly because proper biochemistry was not done. The authors mentioned that the drop in GFP synthesis could be due to interference with translation which is true. However these HigB toxins are all ribosome-dependent, without ribosomes they are all inactive, HigB2 and 3 could just “look” inactive because they have lower affinity for ribosomes but once loaded, may be as active as HigB-TAC. Could the authors comment on this?

-We agree with the reviewer that we cannot exclude that both HigB2 and HigB3 are not toxic because they have a lower affinity for the ribosome. We do not want to push the idea that HigB2 and HigB3 are inactive toxins, especially because our data were obtained in *M. smegmatis* and *in vitro*, and we cannot exclude that certain *M. tuberculosis* specific factors are required for HigB2 and HigB3 to be active. However, the nEMOTE data *in vivo*, the sequence and structural comparisons, and the HigB^{TAC} mutational analysis of possible catalytic residues, strongly suggest that both HigB2 and HigB3 might not be very active RNases, at least when compared to HigB^{TAC}, at least under our conditions. In addition, the fact that both HigB2 and HigB3 behave like the non-toxic HigB^{TAC}[K95A] mutant in the GFP synthesis inhibition assay *in vitro* is in line with such hypothesis.

To further illustrate such differences and for clarity reason, we have included a structural superimposition of the model structures of HigB2 and HigB3 with our structure of HigB^{TAC} and highlighted with color codes the residues of HigB2 and HigB3 that correspond to the important residues of HigB^{TAC} identified in this work (**Supplementary Fig. 10 a and b**). This analysis shows significant sequence and structure variations in (i) the catalytic center residues, (ii) the tRNA-binding residues of the c-terminal helix and (iii) the ribosome interaction regions, which may contribute to the loss of HigB2 and HigB3 toxicity.

In agreement with the reviewer’s comment, we have introduced the possibility that HigB2 and HigB3 could still be active under certain circumstances, including the fact that they might have a lower affinity for the ribosome.

-Results section line 123:

“Note that at high concentration of toxins (*i.e.*, 20 times higher than the one at which HigB^{TAC} wild-type efficiently inhibits translation) we do observe a decrease in GFP synthesis in the presence of HigB2, HigB3, or HigB^{TAC}[K95A]. In contrast, addition of unrelated proteins at the same concentration did not show such a marked effect on GFP synthesis (**Supplementary Fig. 1d**). **This suggests that these toxins might have a lower affinity for the ribosome or be catalytically inactive but still able to interact with the ribosome and, somehow interfere with translation under our *in vitro* conditions.**”

-Discussion section line 326:

“In contrast with HigB^{TAC}, no toxicity was detected when HigB2 and HigB3 were expressed in *M. smegmatis* or in *E. coli*. This suggests that both proteins could have a lower affinity for the ribosome or have lost their toxic RNase activity. Although we cannot exclude that these proteins might still be active

RNases and toxic when expressed in *M. tuberculosis* in the absence of their endogenous antitoxins, the lack of toxicity of both toxins is supported by the fact that several key active residues of HigB^{TAC} (and of other closely related RelE-like toxins) are missing in both HigB2 and HigB3, including residues K58, N73, F93 and K99 of the catalytic center, and the newly identified K113 and R117 residues of the long C-terminal helix α 3 contacting the initiator tRNA (**Supplementary Fig. 10a**). In support of this, the structure of HigB^{TAC} clearly differs from the alpha-fold models of HigB2 and HigB3 in particular in the two loops between β 2- β 3 and β 4- α 3, which contain most of the catalytic residues (**Supplementary Fig. 10b**). Moreover, the long C-terminal helix α 3 of both HigB2 and HigB3, which corresponds to the α 3 of HigB^{TAC}, is negatively charged and may thus not be able to engage interaction with the P-site tRNA (**Supplementary Fig. 10c**). HigB2 and HigB3 also lack the large positive patch that allows the binding of HigB^{TAC} to the 16S rRNA (**Supplementary Fig. 10c**) and the hydrophobic pocket shown to host the mRNA is less well defined (**Supplementary Fig. 10d**). These data suggest that HigB2 and HigB3 may neither be able to efficiently bind translating ribosomes nor be very active as RNases.”

Minor comments:

- Page 3, the authors mentioned that the functions of TA modules remain largely unknown, but then seemingly contradict themselves. This type of sentences have dominated the narrative in papers involving TA modules as a way to avoid recent issues regarding TA functions. While a unique overarching function has been strongly challenged, the function of TAs in plasmid maintenance and more recently in bacterial defence as abortive systems or in phages is well documented (particularly relevant for mycobacterium and demonstrated by Dedrick et al. Nature Microbiology 2017). So this part should be amended accordingly.

-We agree with the reviewer that the role of TA systems in plasmid maintenance and in defense against phages is more and more documented. Therefore, as suggested, we have removed the statement that TA systems functions remain largely unknown from page 3 (and from the abstract) and added the reference to Dedrick et al., 2017. Nevertheless, some works also suggest that certain chromosomal TA systems could contribute to other cellular processes, which we also would like to acknowledge in this brief introduction.

Page 3, line 44:” ...TA systems have been involved in the maintenance of chromosomes, plasmids or other genetic mobile elements, in the defense against phage infection through a process known as abortive-infection and in some cases, in antibiotic persistence *in vivo* in infectious models and in bacterial virulence^{14–20}.”

The main part and methods section lacks any description about the purification of ribosomes or details on how the ribosome-HigB complex was assembled and processed to be used in the grids (only that HigB was used in “high excess” which is completely meaningless). This contrast with the very detailed section on image processing and a major oversight (the authors don’t even mentioned the source of ribosomes, which is very relevant in this case). This should be explained in details in the final version of the paper otherwise it would be very difficult to repeat this structure or this protocol for a similar toxin.

-We thank the Reviewer for pointing out this omission. A new paragraph describing the ribosome purification and details about complex formation was added accordingly.

The following text has been added in Methods section line 616:

“Sample preparation for cryo-EM

For the cryo-EM complex, ribosomes were purified from *E. coli* MG1655. When the culture reached an OD₆₀₀ of 0.8, cells were pelleted, resuspended in FP buffer (20 mM Tris-HCl pH 7.5, 50 mM MgOAc, 100 mM NH₄Cl, 0.5 mM EDTA and 1 mM DTT) and lysed in a French press. The lysate was then clarified by centrifugation at 20,000 × *g* for 45 min at 4 °C. Next, the supernatant was layered 1:1 (v:v) over a high-salt sucrose cushion buffer (10 mM Tris-HCl pH 7.5, 10 mM MgOAc, 500 mM NH₄Cl, 0.5 mM EDTA, 1.1 M sucrose and 1 mM DTT). After ultracentrifugation at 92,000 × *g* for 20 h at 4 °C, the resulting ribosome pellets were resuspended in 1 mL of ‘Ribo_A’ buffer (10 mM Tris-HCl pH 7.5, 10 mM MgCl₂, 50 mM NH₄Cl, 0.5 mM EDTA and 1 mM DTT). To isolate the 70S ribosomes from 30S and 50S ribosomal subunits, the ribosomes were centrifuged at 95,000 × *g* for 18 h at 4 °C through a 10–45% (w/w) linear sucrose gradient in Ribo_A buffer. Gradients were fractionated before determining the A₂₆₀ absorbance profiles. Fractions corresponding to the 70S peak were mixed and diluted in Ribo_A buffer for a final ultracentrifugation at 92,000 × *g* for 20 h at 4 °C. The ribosomal pellets were resuspended in Ribo_A buffer, and flash frozen and stored at –80 °C. To prepare the complex, 25 pmol of fMet-tRNA^{fMet} (VWR, Ref. ICNA0219915410) was first refolded for 2 min at 80 °C in “Buffer I” (10 mM HEPES-KOH pH 7.5, 25 mM MgCl₂, 25 mM, and 20 mM NH₄Cl), and this was followed by a second incubation at room temperature for 30 min. Next, purified 70S ribosomes (12.5 pmoles) were incubated at 37 °C for 15 min in “Buffer-III” (10 mM MgOAc, 10 mM NH₄Cl, 50 mM KCl, 5 mM HEPES-KOH pH 7.5, and 1 mM DTT) with 25 pmoles of *cspA* mRNA and 25 pmoles of the folded fMet-tRNA^{fMet}. Finally, 1,250 pmoles of HigB^{TAC} [K95A] toxin were added, and this was incubated at 37 °C for 5 min. After adjusting concentrations to 160 nM in buffer-III, samples were directly applied to glow-discharge holey carbon films (Quantifoil 3.5/1 μm). These grids were flash-frozen in liquid ethane using a Vitrobot Mark III (FEI).”

-Note that exploitable cryo-EM were obtained only when HigB was used at high concentration when compared to ribosomes. The actual exact proportion used in our experiment are 1 ribosome for 2 mRNAs, 2 tRNAs and 100 HigB^{TAC} [K95A] toxin). See Methods section line 635 “Next, purified **70S ribosomes (12.5 pmoles)** were incubated at 37 °C for 15 min in “Buffer-III” (10 mM MgOAc, 10 mM NH₄Cl, 50 mM KCl, 5 mM HEPES-KOH pH 7.5, and 1 mM DTT) with **25 pmoles of *cspA* mRNA** and 25 pmoles of the folded fMet-tRNA^{fMet}. Finally, **1,250 pmoles of HigB^{TAC} [K95A] toxin** were added, and this was incubated at 37 °C for 5 min.”

To illustrate such conditions and avoid using “high excess”, we have now added this information and a reference to the Methods section in the results part line 207 “Note that exploitable cryo-EM data were obtained only when **high concentration of inactive toxin over ribosomes was used in the presence of full-length *cspA* mRNA (100 toxins for 1 ribosome, 2 tRNAs and 2 mRNAs; see Methods section).**”

It was not justified or elaborated why 7-fold was considered as threshold to choose a cleavage site.

-The reviewer points to the thresholds applied for selecting the candidate cleavage sites of the toxins. The chosen threshold requires seven independent cuts at a given position in at least two replicates. It is based on previous nEMOTE experiments (Refs: Redder, P. *Mapping 5'-Ends and Their Phosphorylation State With EMOTE, TSS-EMOTE, and nEMOTE. Methods Enzymol* 612, 361–391 (2018).43; Kirkpatrick, C. L. *et al. Growth control switch by a DNA-damage-inducible toxin-antitoxin system in Caulobacter*

crescentus. *Nat Microbiol* 1, 16008 (2016); Sierra, R. et al. *Insights into the global effect on Staphylococcus aureus growth arrest by induction of the endoribonuclease MazF toxin*. *Nucleic Acids Res* 48, 8545–8561 (2020); cited in the manuscript) and is usually applied for reducing the background noise. However, the most important criteria remains the simultaneous absence of cuts in all three negative controls, which was enforced in this experiment (see M&M Bioinformatics analyses pp503-513). We are confident that these combined thresholds are sufficiently stringent to reveal toxin targets avoiding potential false positives.

-We have now referred to the Methods section Bioinformatic analysis in the results part line 143.

In general the authors tend to abuse terms such “significant, high, low, likely”. They should be precise.

-As requested, we went throughout the manuscript and reduced the use of these terms.

When comparing structures, please provide structural alignments, otherwise interpreting results becomes very difficult, specially when discussing substitutions between different TA toxins (for example in Fig 6b). Without structural alignments is for example very hard to interpret and follow why HigB2 and HigB3 are not toxic based on the the relevant residues and substitutions in HigB-TAC. This point was already raised by other reviewers and the current Fig are still not great in this regard.

Structure-based sequence alignment are provided in Supplementary Fig. 6b and new Supplementary Fig. 10a, and Structure superimposition are now shown in Supplementary Fig. Fig 6a, new Supplementary Fig. 10b and new Supplementary Fig. 9.

-Fig. 6b focus on how the different toxins interact in the ribosome A-Site with the mRNA and the P-Site tRNA, the structure of the different toxin were all aligned on the P-site tRNA using UCSF chimera. To avoid overcrowding and to facilitate interpretation we have added a new Supplementary Fig. 9, that shows a larger version of the aligned structures of Fig. 6b and this time with the structure of HigB^{TAC} [K95A] (red) superimposed.

-Supplementary Fig. 6. The structure of the HigBTAC [K95A], *S. pneumoniae* HigB (PDB 6AF4), *V. cholera* HigB2 (PDB 5JA9), and *E. coli* RelE [R45A-R81A] (PDB 4V7J) were already structurally aligned with UCSF chimera but not superimposed. To facilitate the interpretation we proposed a new version of the figure with HigBTAC [K95A] (transparent white) superimposed to the other toxins (new Supplementary Fig. 9).

-Supplementary Fig. 7. To focus on how the different toxin interact with the DC, the structure of the different toxin interacting with the ribosome were aligned on the 16S mRNA using UCSF chimera. For clarity the structure could not be superimposed but the caption was modified accordingly.

-Supplementary Fig. 10. The AlphaFold models of Mtb HigB2 and HigB3 are structurally aligned on the cryo-EM structure of HigBTAC [K95A], using STAMP and colored based on the per-residue RMSD with HigBTAC from blue (low RMSD *i.e.*, small structural difference) to red (High RMSD *i.e.*, large structural difference). However in agreement with the reviewer and to facilitate the interpretation we made a new version of the figure with HigB^{TAC} [K95A] (transparent white) superimposed to the other toxins and modified the caption accordingly.

The authors also used unnecessarily the overlay of electron densities in the figures. While it is good they showed it perhaps once like in Fig. 4 to show that HigB is indeed bound, in the rest of the figures that convey mechanistic information, this only difficult the interpretation and should be removed or shown as supplementary, keeping the main figures without the densities (this is critical in Fig 5).

-As suggested by the reviewer, we have removed the electron densities from Fig. 5 and kept it in Fig. 4. We kept the electron density only in the close-up view in Fig. 6a as it prove that the experimental data are sufficiently resolved to build an accurate model of the interaction between HigB^{TAC} C-ter helix and the tRNA (including residues side chain, as previously discussed by Reviewer 3).